

# Long range transport of Canadian Wildfire smoke to Europe in Fall 2023: aerosol properties and spectral features of smoke particles

Akriti Masoom[1,a], Stelios Kazadzis[1], Robin Lewis Modini[2], Martin Gysel-Beer[2], Julian Gröbner[1], Martine Collaud Coen[3], Francisco Navas-Guzman[3,4], Natalia Kouremeti[1], Benjamin Tobias Brem[2], Nora Kristina Nowak[2], Giovanni Martucci[3], Maxime Hervo[3], Sophie Erb[3,5]

[1]Physikalisch-Meteorologisches Observatorium Davos, World Radiation Center, Davos, 7260, Switzerland
[2]PSI Center for Energy and Environmental Sciences, 5232 Villigen PSI, Switzerland
[3]Federal Office of Meteorology and Climatology, MeteoSwiss, Payerne, 1530, Switzerland
[4]Applied Physics Department, University of Granada, Granada, 18071, Spain
[5]Environmental Remote Sensing Laboratory, EPFL, Lausanne, 1015, Switzerland
[a] now at: Oak Ridge Associated Universities, Oak Ridge, 37830, United States and Goddard Space Flight Center, Greenbelt, 20771, United States

*Correspondence to*: Akriti Masoom (akriti.masoom@nasa.gov)

**Abstract.** Canadian wildfires in Fall of 2023 were unique regarding the observed aerosol properties using remote sensing and in situ measurements for both short-range and long-range transported plumes across North America and Europe. One of the highlights was the observation of special concave spectral curvature in AOD having maxima at higher wavelengths than the minimum measured wavelength which led to negative values of Ångström exponent in spectral ranges below 500 nm. Along with this, large accumulation mode size distributions with volume median diameters reaching about 800 nm were observed. Another unique observation was the non-monotonic spectral curvature in single scattering albedo (SSA). For most of the stations, SSA increased in the UV-Visible region and/or further remaining either constant or decreasing at longer wavelengths as observed from retrievals from column integrated (AOD) from remote sensing and coefficients from in situ measurements. Additionally, two stations in Canada and one in Europe were found to have a well-defined peak in AOD at 500 nm. These Canadian stations also displayed a non-monotonic spectral SSA with maxima at 675 nm, while the high altitude stations of Europe showed monotonically increasing SSA. Finally, a much higher (approximately 5 times) UV absorption than visible absorption indicated the presence of brown carbon and/or tar balls, which have a strong spectral dependence in imaginary refractive index. The SSA concave spectral curvature denotes the mix of black carbon and non-absorbing particulate matter and influence of particle size, while the AOD concave spectral curvature is attributed to particle size.

## 1 Introduction

Fire weather enhancement and wildfire activities are increasing as a consequence of ongoing climate change with more frequent heatwaves and intensified drought seasons (Abatzoglou et. al., 2019; Perkins-Kirkpatrick and Lewis 2020; Weilnhammer et. al., 2021; Fischer et. al., 2021; UN, 2022). Forests play a crucial role in carbon exchanges between the



atmosphere and biosphere leading to carbon sequestration of approximately 30% of global carbon emissions annually from anthropogenic sources (Chen et. al., 2024). However, with increasing wildfire activities, forests are becoming carbon sources rather than being a designated carbon sink. Wildfires also significantly influence solar radiation altering the Earth-Atmosphere

radiative balance. Smoke particle absorption and scattering lead to a reduction in surface solar irradiance (e.g., Corwin et. al., 2025; Masoom et al., 2023), while simultaneously increasing atmospheric heating due to absorption by black carbon (Jacobson, 2001). These changes can affect local weather patterns but also photovoltaic energy generation (Zhao et al., 2021). Furthermore, the long-range transport of smoke aerosols can lead to regional and even global impacts on radiation budgets, cloud formation, and climate dynamics (Peterson et al., 2018).

The boreal regions are experiencing an increased annual wildfire activity (Balshi et. al., 2009; Flannigan et. al., 2009) as a consequence of the amplified warming due to climate change and increase in droughts in this region. The resulting large increase in the burnt area over the recent time period in Canada and Alaska (Calef et. al., 2015; Hanes et. al., 2019) is expected to further increase in future (Amiro et. al., 2009; Flannigan et. al., 2005; Park et. al., 2023; Lund et. al., 2023; Allen et. al., 2024). Carbon emissions associated with wildfire activity in the boreal forest regions depend strongly on the fuel availability

and type (Walker et. al., 2020; Allen et. al., 2024).

Wildfire smoke transport from North America to Europe is not rare and has been reported by many studies (and the references therein) some of which are discussed here. Baars et al. (2021) studied the September 2020 California wildfires which produced large amounts of smoke comparable to moderate volcanic eruptions that was lifted into the free troposphere (Peterson et al., 2018) and was transported within 3–4 days from the west coast of US to central Europe. Another study by Zheng et al. (2020)

also supported the possibility of North American boreal wildfire smoke plume having elevated injection heights and longer wildfire aerosol lifetime for the record-breaking Canadian wildfires in August 2017 that led to prominent ageing that requires understanding and quantification of their effects on radiation and climate. These aged wildfire aerosols transported over North Atlantic Ocean over a period of approximately 2 weeks had high single scattering albedos and low absorption Ångström exponents as compared to fresh/slightly aged smoke. Another study by Baars et al., (2019), showed stratospheric perturbation

caused by wildfire smoke owing to strong thunderstorm–pyrocumulonimbus activity associated with wildfires in western Canada in 2017 that spread over the entire Northern Hemisphere. The transport of the plume was detected by European lidar stations all over Europe with smoke layer observation at height above 15 km. The stratospheric aerosol optical thickness at 532 nm was observed to be significantly above the stratospheric background values.

Sicard et al. (2019) studied the 2017 wildfires in Canada/United States mostly associated with temperate coniferous forests

and analyzed the long range transported smoke particles over the Iberian Peninsula observing aerosol optical depth (AOD) at 440 nm up to 0.62, Ångström exponent (AE, referring to extinction AE) of 1.6–1.7 signifying dominance of small particles with fine mode fraction above 0.88 and low absorption AOD at 440 nm (<0.008) and large single scattering albedo at 440 nm (>0.98). They also showed that a slow travelling smoke plume layer rapidly (approximately 1 day) reached upper troposphere and lower stratosphere with large scale horizontal dispersion. While another smoke layer reached the upper troposphere

comparatively at a slower pace (approximately 2.5 days) possibly due to entrainment by strong subtropical jets highlighting



the importance of horizontal as well as vertical transport during wildfire events. Another study by Ohneiser et al. (2023) on tropospheric and stratospheric lofting of smoke due to radiative heating as a result of high absorption of sunlight by optically thick smoke layers leading to self-lofting of smoke plume from injection heights in lower and middle free troposphere to tropopause in the absence of pyrocumulonimbus convection. This study found that the lofting rates are dependent on the

aerosol optical thickness, aerosol layer height, aerosol layer depth as well as black carbon fraction and that self-lofting contributes to the vertical transport of smoke.

A study by Ceamanos et. al. (2023) also reported transported wildfire plume thousands of kilometres away to Europe from wildfires burning in Western US in the summer of 2020. This study mentioned the usefulness of satellite and model data in monitoring such transports and also disagreements during intense smoke activity that might be due to biases in satellite retrieval

algorithms or differences in overpasses time of satellites as compared to assimilation window time in models or fire emission estimation limitations in models. Another study in this direction by Shang et. al. (2024) on transport of Alberta, Canada wildfires in May–June 2019 to Europe, found that smoke aerosol amounts in model simulations were consistently lower as compared to satellite retrievals which highlight reanalysis model limitations in reproducing smoke properties.

The 2023 Canadian wildfires were unique in some aspect as reported by several studies so far as discussed here. There were

some factors that contributed towards the 2023 Canadian forest fires being extreme in scale and intensity which includes the fact that the average annual area burned was more than seven times in comparison to the preceding four decades (Whitman et. al., 2024). According to a study by Byrne et. al., 2024, the carbon emission magnitude was 647 TgC (570–727 TgC) which is comparable to the annual emissions from fossil fuel by large nations (Friedlingstein et. al., 2022) and the principal driver for the spread of fire being widespread hot and dry weather as 2023 has been recorded as the driest and warmest year since at least

1980. Another study by MacCarthy et. al. (2024) reported that the 2023 Canadian wildfires were record-breaking as a consequence of extreme heat and low rainfall due to climate change, with a burned area of approximately 7.8 million hectares, accounting for more than a quarter of global tree cover loss in 2023 leading to emission of approximately 3 billion tons of $CO_2$. Apart from unprecedented quantities of $CO_2$ released in the atmosphere, approximately 0.14 Pg $CO_2$ equivalent of other greenhouse gases (GHG) including $CH_4$ and $N_2O$ were also emitted by August 31 according to a study by Wang et. al. (2024).

The study also reported that this Canadian wildfire also impacted many areas due to long-range transport in the northern hemisphere, contributing severely to PM2.5 pollution in the north-eastern United States and north-western China by up to 2 $\mu g\ m^{-3}$.

A study by Jain et. al. (2024) looked into the probable causes of the 2023 wildfire season in Canada that spanned between mid-April to late October. They found that it can be attributed to several environmental causes including early snowmelt,

multiannual drought conditions in western Canada, and the rapid transition to drought in eastern Canada. Also, the anthropogenic climate change enabled sustained extreme fire weather conditions and the mean May–October temperature over Canada was 2.2° C warmer in 2023 as compared to the average between 1991–2020. Apart from setting new records, the 2023 Canadian wildfire season profoundly impacted environment, weather, wildlife as well as human settlements. This resulted into



evacuation of more than 200 communities with hazardous smoke aerosol exposure to millions and increased unmatched fire-
fighting resource demands (Jain et. al., 2024) highlighting the increasing wildfires related challenges.

However, most of the studies that have been published so far on 2023 Canadian wildfires (to the best of the authors knowledge)
are mainly focused on the summer wildfire season. However, the wildfires of Fall 2023 were unique in a way that it revealed
special characteristics in spectral AOD variations as well as in other associated aerosol properties which is the central part of
investigation of this manuscript.

There is an observed concave shape associated with a fictitious AOD diurnal cycle which is a function of inverse of the airmass
and therefore the largest magnitude of this diurnal variation is observed at midday (Cachorro et al., 2008). AOD displays this
fictitious diurnal cycle which is an artifact resulting from the presence of an incorrect calibration constant (or an equivalent
effect, such as filter degradation or electronic instability) (Cachorro et al., 2008, Giles et al., 2018, Xun et al., 2021). This
fictitious AOD diurnal cycle is prominent for UV wavelengths calibration due to the sensitivity of these channels to larger
calibration error (Cachorro et al., 2008, Slusser et al., 2000). However, the concave AOD variation presented in the current
manuscript is not associated with the fictitious diurnal cycle but is related to the spectral variation, hence referred to as concave
spectral curvature in AOD. This kind of spectral AOD variation was observed in Eck et al. (2023) showing peak in AOD at
500 nm associated with California/Oregon fires of 2020 and an extreme size distributions retrievals for long range smoke
plumes transport. An accumulation mode number mean diameter of 0.34 µm was observed by Fiebig et al. (2003) for particle
size distributions in an airborne in situ measurements for forest fire plume transport in 1998 from Northern Canada. The authors
observed this as possibly the largest observed value for size distribution in a forest fire plume. However, there was not no
spectral optical properties measurements available for this plume transport.

The current manuscript presents the spectral optical properties associated with such extreme size distribution as well as a
synergistic approach of aerosol remote sensing and in situ measurements to further explore this phenomenon. Moreover,
previous studies have demonstrated the advantages of combining active and passive remote sensing to characterize the optical
and microphysical properties of fresh smoke aerosols close to their sources (Alados-Arboledas et al., 2011). This manuscript
utilizes remote sensing and in situ measurements to discusses the probable reason for the peak in AOD at 500 nm at three
stations with different geographical and meteorological conditions being an example of two being near the source having high
intensity of smoke plume and short-range transport while the other being a long-range transport case with a more prominent
ageing of the plume. Apart from this, the manuscript also presents a comparison with satellite observations and model
reanalysis dataset in this plume transport case.



## 2 Data and Methodology

### 2.1 Aerosol property measurements

#### 2.1.1 Columnar aerosol property measurements

The Aerosol Robotic Network (AERONET) uses Cimel sunphotometers (AERONET, 2024) and has a centralized data processing and distribution system providing instrument calibrations and standardized data acquisition to retrieve aerosol optical, microphysical, and radiative properties through ground-based passive remote sensing. AERONET Cimel sun photometers are calibrated at the high-altitude station in Izaña by the Langley plot method (Holben et al., 1998). The direct-sun algorithm of AERONET, using the version 3 processing algorithm (Giles et al., 2019) employed in this work, includes

level 1.5 AOD measurements at 340, 380, 440, 500, 675, 870 and 1020 nm and AE retrievals at 440–870, 380-500, 440-675, 500-870 and 340–440 nm. Level 1.5 AERONET data have been utilized for several stations for tracing the plume as presented in Section 3.2. AERONET inversion products from level 1.5 have also been utilised in this work including single scattering albedo (SSA), absorption aerosol optical depth (AAOD), absorption Ångström exponent (AAE), volume size distribution (VSD) and refractive indices (RI).

Precision filter radiometers (Wehrli in WMO, 2005), part of the Global Atmospheric Watch-Precision Filter Radiometer (GAW-PFR) network, perform aerosol optical depth measurements at four wavelengths namely 368, 412, 500, and 862 nm (Wehrli, 2000). The primary calibration is performed using the reference triads of PFRs which are calibrated regularly at the high-altitude stations of Mauna Loa in Hawaii, USA and Izaña in Tenerife, Spain using the Langley calibration method (Nyeki et. al., 2015). Instruments are calibrated yearly or in 2 years on the basis of instrumental and logistic aspects and the calibration

uncertainty is assured to be within ±1% (Kazadzis et al., 2018a). Quality-assured AOD data are obtained after pre- and post-deployment calibrations. Post-deployment calibrations are done either by the World Optical depth Research and Calibration Center (WORCC) issued calibration certificates or by Langley site calibrations (Kazadzis et al., 2018a, Toledano et al., 2018). For this analysis, we have used the PFR AOD and AE data for DAV (46.80° N, 9.80° E, 1580 m above sea level (asl)) and JFJ (46.32° N, 7.59° E, 3580 m asl). AERONET/CIMEL and PFR instruments have been repeatedly intercompared in various

studies in order to ensure homogeneous retrievals (e.g., Kazadzis et al., 2018b, Karanikolas et al., 2024).

In addition to the sun photometers and filter radiometers as mentioned above, spectral measurements have also been used. The transportable reference spectroradiometer QASUME consists of a scanning double monochromator with a full width at half maximum (FWHM) of 0.86 nm and measures over the spectral wavelengths from 280 nm to 550 nm. The whole system resides in a temperature-controlled enclosure to allow outdoor operation under varying ambient conditions. The solar radiation is

collected with a temperature stabilised diffuser connected via an optical fiber to the entrance slit of the monochromator. A portable lamp monitoring system allows for the calibration of the whole system while being deployed in the field (Gröbner 2017, Hülsen, et al., 2016). A collimator tube with a full opening angle of 2.5° is mounted on an optical tracker to which the diffuser head can be fitted, allowing the measurement of direct solar spectral irradiance. The measurements of direct solar



spectral irradiance have a standard relative uncertainty of less than 1%, resulting in a standard uncertainty of 0.014 at 310 nm
to 0.007 at longer wavelengths in AOD at an airmass of 1.5 (Gröbner 2023).

The BiTec Sensor (BTS) instruments, manufactured by Gigahertz Optik GmbH, are a system composed of two array-spectroradiometers. The spectral range from about 320~nm to 1000~nm is covered with a 2048-pixel Si BTS2048-VL-TEC-WP with a nominal spectral resolution (FWHM) of 2.5~nm whose characterization was described in Zuber et al. (2018a, b), while the spectral range from 1000~nm to 2150~nm is measured with a BTS2048-IR-WP with a nominal spectral resolution
of 8~nm (FWHM) with 512 pixel and an extended InGaAs detector. Each spectroradiometer has a collimator to measure direct solar spectral irradiance. The instruments used in this analysis were calibrated at the Physikalisch-Meteorologisches Observatorium Davos and World Radiation Center using the same portable lamp system as described above for QASUME, resulting in similar uncertainties in spectral irradiance and AOD as for QASUME (Gröbner 2023).

### 2.1.2 Aerosol in situ measurements

Smoke plume optical, microphysical and chemical particle properties were analyzed using the comprehensive in situ instrumentation at the High-Altitude Research Station JFJ. The aerosol measurements of the JFJ site are part of the GAW program, the pan-European Aerosol, Clouds and Trace Gases Research Infrastructure (ACTRIS) and the Swiss National Air Pollution Monitoring Network (NABEL). This instrumentation is briefly described here.

Aerosol light scattering was measured with an integrating Nephelometer (TSI 3563) which provides the total scattering and
backscattering coefficients at 450, 550 and 700 nm wavelengths. The instrument output was corrected for angular truncation using the network recommendations (ACTRIS-CAIS-ECAC, 2024), which rely on the scheme from Anderson and Ogren (1998). Light absorption coefficients at seven wavelengths (370, 470, 520, 590, 660, 880 and 950 nm) were determined by a dual spot Aethalometer (MAGEE scientific AE33). The network recommended instrument data processing algorithms based on the work of Drinovec et al. (2015) were applied to correct for non-idealities in filter attenuation-based instrument.

In addition to optical properties, the particle size distributions ranging from 0.01 to 0.8 µm and 0.5 to 20 µm, were measured by a mobility particle size spectrometer (Wiedensohler et al., 2012), an aerodynamic particle sizer (APS, TSI 3321), and a white light optical particle sizer (Palas Fidas® 100), respectively. Merged particle size distributions covering the diameter range from 0.01 to 10 µm were generated by joining the MPSS size distributions below 0.6 µm with the averages of the APS and Fidas size distributions above 0.6 µm (after the APS measured diameters were shifted from aerodynamic to mobility
diameters assuming an effective density of 1.6 g cm$^{-3}$).

A condensation particle counter (TSI 3772), with a lower cut-off size of 10 nm provided the total particle number concentrations.

The chemical composition of the non-refractory particulate matter, with aerodynamic diameters smaller than 1 µm was further analyzed in situ and in real time with a Time-of-Flight Aerosol Chemical Speciation Monitor (TOF-ACSM, Aerodyne Inc.)
that provides the linear detection of sulphate, nitrate, ammonium, chloride and organic aerosol species through a two-step



thermal vaporization (approximately 600 C) and electron impact ionization process. More details on this instrument can be found in Fröhlich et al., (2013 and 2015).

## 2.2 Aerosol vertical profiles

The RAman Lidar for Meteorological Observations (RALMO) was designed by MeteoSwiss and the École Polytechnique Fédérale de Lausanne (EPFL) and is operated at the MeteoSwiss station of Payerne (PAY; 46.80° N, 6.93° E, 492 m asl), Switzerland, since the year 2007. It provides, continuous operational measurements of humidity since 2008 and temperature and aerosol backscatter since 2010. RALMO is fully automated and operates continuously except in the presence of precipitation or a cloud ceiling below 1000 m above ground level. Data are processed in near-real-time and made available to the MeteoSwiss database and to the international database of NDACC, GRUAN and EARLINET. RALMO uses a narrow field-of-view, narrowband configuration, a UV laser at 355 nm, and four telescopes with 30-cm diameter, fiber-coupled to two grating polychromators for the retrieval of water vapour, temperature and aerosol backscatter. The optical design of RALMO allows the retrieval of the water vapor and the temperature within the troposphere up to the tropopause, while the aerosol backscatter can be retrieved up to the middle stratosphere allowing the detection of elevated stratospheric aerosol plumes. A more detailed description of the aerosol, temperature and humidity transceiver systems is provided in Martucci et al. (2021), and Dinoev et al. (2013) and Brocard et al (2013), respectively.

CHM15k ceilometer data has been used for vertical profile observations at Davos which is a one-wavelength backscatter lidar at 1064 nm. The vertical profile dataset is obtained from V-Profiles (https://vprofiles.met.no/about/; last accessed: May 25, 2025) including the attenuated backscatter signal at 1064 nm from 0 to 6000 m above ground level.

## 2.3 Satellite observations

Satellite observation of AOD was obtained from the daily retrievals of the MODerate resolution Imaging Spectroradiometer (MODIS) Collection 6.1. MODIS Level 2 AOD retrievals at 550 nm (Levy et al., 2013; Wei et al., 2019b) from the Dark Target (DT) retrieval algorithm provide AOD values above ocean and land while the Deep Blue (DB) retrieval algorithms provides AOD values above land.

MODIS AOD values were considered for the analysis period between 20 September 2023 and 05 October 2023 as is presented in Section 3.3. During the same time period, MODIS true colour images were also considered for tracing of the plume as presented in Fig. A2 in Appendix.

## 2.4 Aerosol trajectory modelling and model reanalysis

Aerosol source and transport monitoring was performed using Hybrid Single-Particle Lagrangian Integrated Trajectory (HYSPLIT) model that uses a hybrid of Lagrangian and Eulerian approaches (Stein et al., 2015).



For analyzing the source and transport of the wildfire plumes, the HYSPLIT is used over the regional to global scale to account for the transport of pollutants, their dispersion, and deposition.

In this analysis, 8 days backward trajectories ending at 12:00 UTC at the desired locations were generated using Global Data Assimilation System meteorological data at eight levels between 0.5 and 5 km.

For the AOD dataset from a global model, we considered The Modern-Era Retrospective Analysis for Research and
Applications, Version 2 (MERRA2) which is an atmospheric reanalysis product of Global Modeling and Assimilation Office of National Aeronautics and Space Administration (Gelaro et al., 2017) that includes assimilation of aerosol observations, several improvements to the representation of the stratosphere including ozone, and improved representations of cryospheric processes.

We specifically considered the AOD at 550 nm from this dataset for the time period of the analysis from 20 September 2023
to 05 October 2023, as presented in Section 3.3.

## 3 Results

### 3.1 Smoke plume detection over Swiss Alps

### 3.1.1 Jungfraujoch, Davos and Payerne

The long-range transported smoke from several wildfires burning in Canada in September 2023 reached the Swiss Alps in late
September (reaching JFJ on 30 September) and beginning of October (reaching DAV and PAY on 01 October) from the northern part of the European mainland after crossing the North Atlantic Ocean (Fig. 1a).

This was further confirmed by the sky-camera images (Fig. 1b) as well as the Ceilometer measurements (Fig. 1c) on 01 October 2023 at DAV, which detected the presence of plumes above 2 km height above ground level, later settling below 2 km during the day of 01 October. The air mass back trajectories at DAV (Fig. 1a) pointed towards the central European aerosol sources
originating from the North-East of Canada, where several wildfires were active during the September of 2023 (Byrne et al., 2024; Jain et al., 2024; Chen et al., 2025). The transported smoke plume was also detected by RALMO in PAY on 01 October 2023 as presented in Fig. 1d with the presence of a thick aerosol layer slowly lowering in altitude from 2 km to 1 km, the aerosol layer is characterized by a total backscattering ratio value exceeding 4 from 06:00 to 18:00 UTC on 01 October.



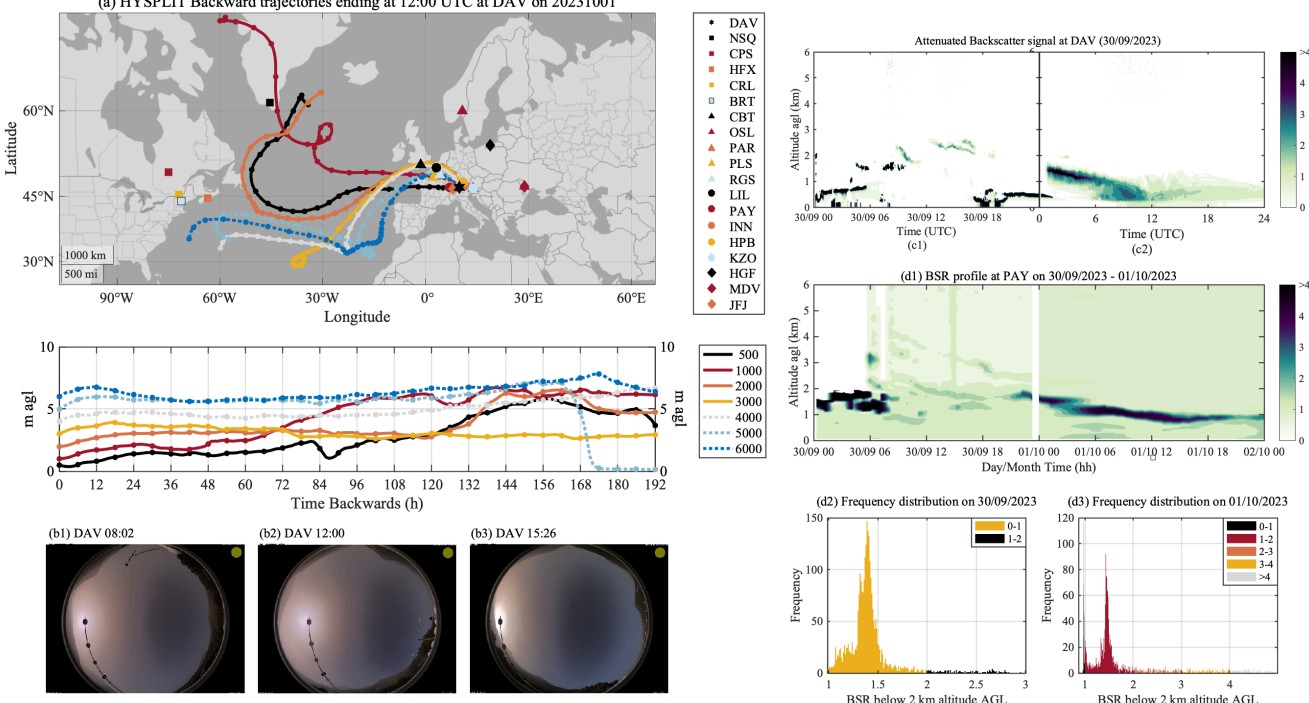

**Figure 1: HYSPLIT (a1) backward, and (a2) forward trajectories ending at and beginning from Davos, respectively, at 12:00 UTC and (b1-b3) sky camera images during the day of 01 October 2023. Vertical profiles of aerosol backscattering from Ceilometer at Davos on (c1-c2) 30 September 2023 and 01 October 2023, respectively. RALMO based (d1) total backscattering ratio (BSR) profile, and frequency distribution on 30 September (d2) and on 01 October (d3) below 5 km altitude with respect to the cluster BSR values.**

### 3.1.2 Anomalous spectral aerosol optical properties at Davos and Jungfraujoch

The event was characterized by increased aerosol load from ground-based aerosol measurements on 01 October 2023, at DAV and on 30 September 2023 at JFJ. Apart from the elevated AOD values, a unique feature of this long-range smoke plume transport event was the spectral AOD dependence at DAV as observed from ground-based measurements (Fig. 2), which was special, as the spectral AOD first increased from 310 nm to approximately 500 nm and then decreased at longer wavelengths i.e., a distinct spectral curvature in spectral AOD variation with maxima around 500 nm could be observed. Figure 2a shows the spectral AOD comparison as captured with multiple filter- and spectro- radiometers on 01 October at 10:00 UTC at DAV that further confirms the relevance of the occurrence and spectral curvature in AOD variation. Similar spectral AOD variation was also observed for remote GAWPFR station of JFJ with the PFR instrument as presented in Fig. 2b that also had spectral curvature in AOD during some time stamps on 30 September 2023. In addition, the spectral AOD analysis for PAY, exhibiting similar spectral behaviour, is provided in Section 3.2.1 with the available AERONET measurements.





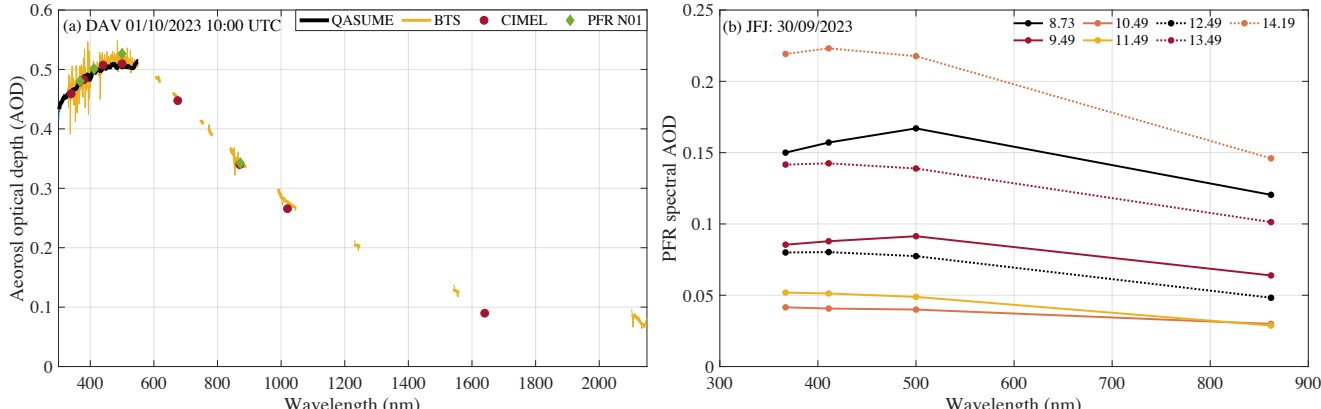

**Figure 2: (a) Instantaneous spectral AOD variation at 10:00 UTC in DAV from sun photometer (Cimel and PFR) and spectroradiometer (BTS and QASUME). (b) Spectral AOD variation at JFJ on 30/09/2023 from PFR calculated as mean with 1 h time stamp for the whole day (time represented in the legend in UTC).**

This unique feature in spectral AOD variation can be considered as an extreme case scenario that was observed for the first time in the measurement history of DAV. The AOD variation at DAV from 2004 to 2022 with nearly 2 decades of AOD values being below 0.11 and the cases of AOD being above 0.5 is only 1% of the total measurements (Figure A1 a-d). Figure A1e shows the peak daily mean spectral AOD values from 2004-2022 at DAV from GAWPFR measurements during the same period. The daily mean spectral AOD values are calculated ensuring that there is valid measurement at all wavelengths and considering AOD values up to 3 precision points. The highlighted spectral AOD variation and peak in AOD at 500 nm, was observed only on one day - 01 October 2023 in nearly two decades of continuous measurements at DAV.

### 3.1.3 Sensitivity of AOD spectral dependence to aerosol particle size distribution, mixing state, refractive index and shape at Jungfraujoch and Payerne

To further investigate this special spectral AOD curvature, we studied the detailed microphysical and chemical properties of the particles and gases measured in situ in the smoke plume during its passage over JFJ on 30 September and 01 October 2023. These measurements are summarized in Fig. 3 and Table A1. The plume was characterized by elevated concentrations of CO and Benzene in the gas phase (approximately 100 – 300 ppb and 20 – 55 ppt, respectively; Fig 3a; note that benzene concentrations were not measured during the plume peak due to the instrument calibration schedules). The mass concentration of particles with diameters less than 1 µm (PM1) peaked at approximately 30 µg m$^{-3}$ during the beginning of the plume period and averaged around 1.3 µg m$^{-3}$ during the remainder of the plume period. These concentrations are drastically greater than the corresponding concentrations before and after passage of the plume (approximately 0.1 – 0.6 µg m$^{-3}$). The particles were mostly composed of organic compounds (approximately 0.7 to 12 µg m$^{-3}$), with only minor contributions from black carbon (approximately 0.04 to 1.4 µg m$^{-3}$) and the inorganic species sulphate ($SO_4^{2-}$; approximately 0.08 to 0.2 µg m$^{-3}$) and nitrate ($NO_3^-$; approximately 0.02 to 0.6 µg m$^{-3}$).





**Figure 3: Time series of gas and aerosol properties measured in situ in the smoke plume as it passed over JFJ on September 30 and October 1, 2023: (a) concentrations of carbon monoxide (CO) and benzene in the gas phase; (b) the mass concentrations of all particles with diameters less than 1 μm (PM1) as well as the organic components (Organics), nitrate (NO$_3$), sulphate (SO$_4$), and black carbon (BC); (c) scattering coefficients 550 nm and absorption coefficients at 370 and 550 nm, and single scattering albedo (SSA) at 550 nm; (d) scattering Ångström exponents (SAE) from 450 – 700 nm, absorption Ångström exponents (AAE) from 370 – 880 nm, and SSA Ångström exponents (SSA_AE) from 450-700 nm; (e) particle number size distributions as an image plot with overlaid white trace indicating the total particle number concentration; and finally (f) particle volume size distributions as an image plot with overlaid white trace indicating the total particle volume concentration. The grey shaded region indicates the period when the plume was impacting the station.**

The smoke particles scattered and absorbed considerable amounts of light. Aerosol scattering and absorption coefficients at 550 nm were up to 366 and 19 Mm$^{-1}$, respectively, resulting in single scattering albedo (SSA) values at 550 nm of around 0.95, which are similar to the corresponding SSA values before and after the plume period. The smoke particles had unique optical exponents: e.g., scattering Ångström exponents (SAE) from 450 – 700 nm were approximately 0, absorption Ångström exponents (AAE) from 370 – 880 nm were approximately 2, and SSA Ångström exponents were approximately -0.10 during the plume period (Fig. 3d). Usually, aged, long range transported wildfire plumes measured at JFJ have AAE values below 1.5



because of bleaching and evaporation of the brown carbon content (Forrister et al., 2015). Such values are closer to the spectral properties of BC-rich fossil fuel combustion plumes that have AAE values approximately 1.0. However, in this special case on September 30 and October 1, 2023, UV absorption (370 nm) was much higher (approximately 5 times) than visible absorption indicating the presence of brown carbon and/or tar balls, which have a strong spectral dependence in the imaginary refractive index (Corbin et al., 2019). The negative SSA Ångström exponents observed for these smoke particles are also remarkably unique. Normally such negative values are only observed at JFJ during Saharan dust events (SDEs; Collaud Coen et al., 2004). During this special event, the unique optical properties of the smoke particles led to a false trigger of the JFJ SDE alert.

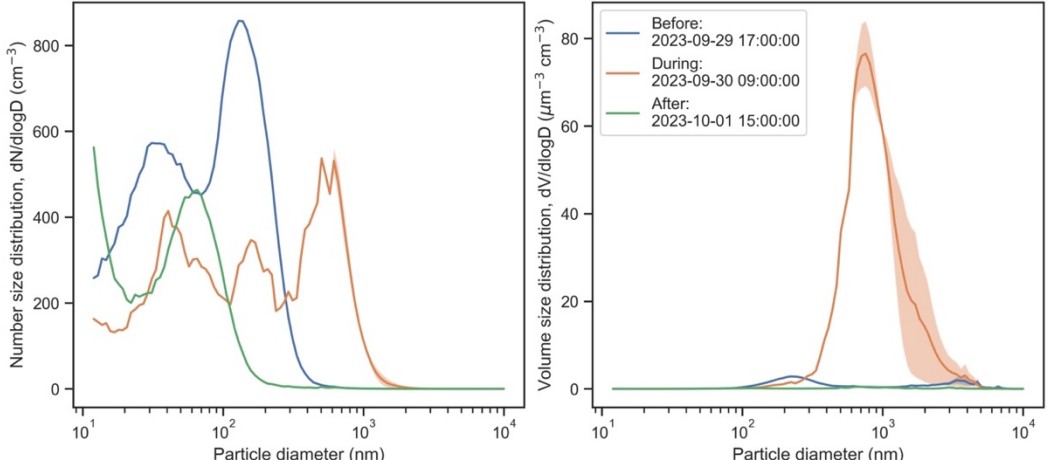

**Figure 4: Particle (a) number and (b) volume size distributions measured in situ at JFJ before, during, and after the passage of the smoke plume.**

In contrast to the highly elevated aerosol mass concentrations in the plume, the number concentrations of particles in the plume were relatively similar to those before and after plume passage (in all cases within a range of $250 - 750$ particles cm$^{-3}$). This observation is explained by the size distributions of the smoke particles, shown as time series in the image plots in Fig. 3e and 3f, and for selected time periods in Fig. 4. The number size distributions of particles below approximately 500 nm diameter before, during, and after the plume contained multiple modes and were similar in magnitude. However, during the plume, an additional size mode centred at around 600 nm appeared in the particle number size distribution (Fig. 4a). This mode is even more apparent when plotting the distributions as a function of aerosol volume or VSD (Fig. 4b), demonstrating that these particles led to the substantial increases in the total aerosol volume (and mass) concentrations. While a typical particle size distribution for a transported and aged wildfire smoke plume shows a shifted peak within the fine mode, the large shift towards unusually large particle diameters observed here is a very rare event, as also reported by Eck et al. (2023) and Fiebig et. al. (2003). To further investigate the observed spectral dependence of AOD of this wildfire plume, we performed theory calculations constrained with observational data. All particles were assumed to be identical homogeneous spheres (except for their diameter), i.e., all components including BC and BrC were assumed to be present as homogeneous internal mixture. The



size distribution was constrained MPSS and APS in situ data, and the complex refractive index was taken from DAV

AERONET retrieval results. Figure 5 shows the spectral dependence of extinction coefficient (Fig. 5b), scattering coefficient (Fig. 5d), and absorption coefficient (Fig. 5f) calculated with Mie theory for the polydisperse aerosol as described above. It is clearly seen that the inverted spectral dependence of extinction, i.e., positive gradient below ~600 nm wavelength, is driven by similar inverted spectral dependence of scattering. By contrast, calculated absorption has a negative gradient across the entire spectral range corresponding to an AAE significantly larger than unity. Given the high SSA of the aerosol sample, it is

not surprising that extinction is primarily driven by scattering.

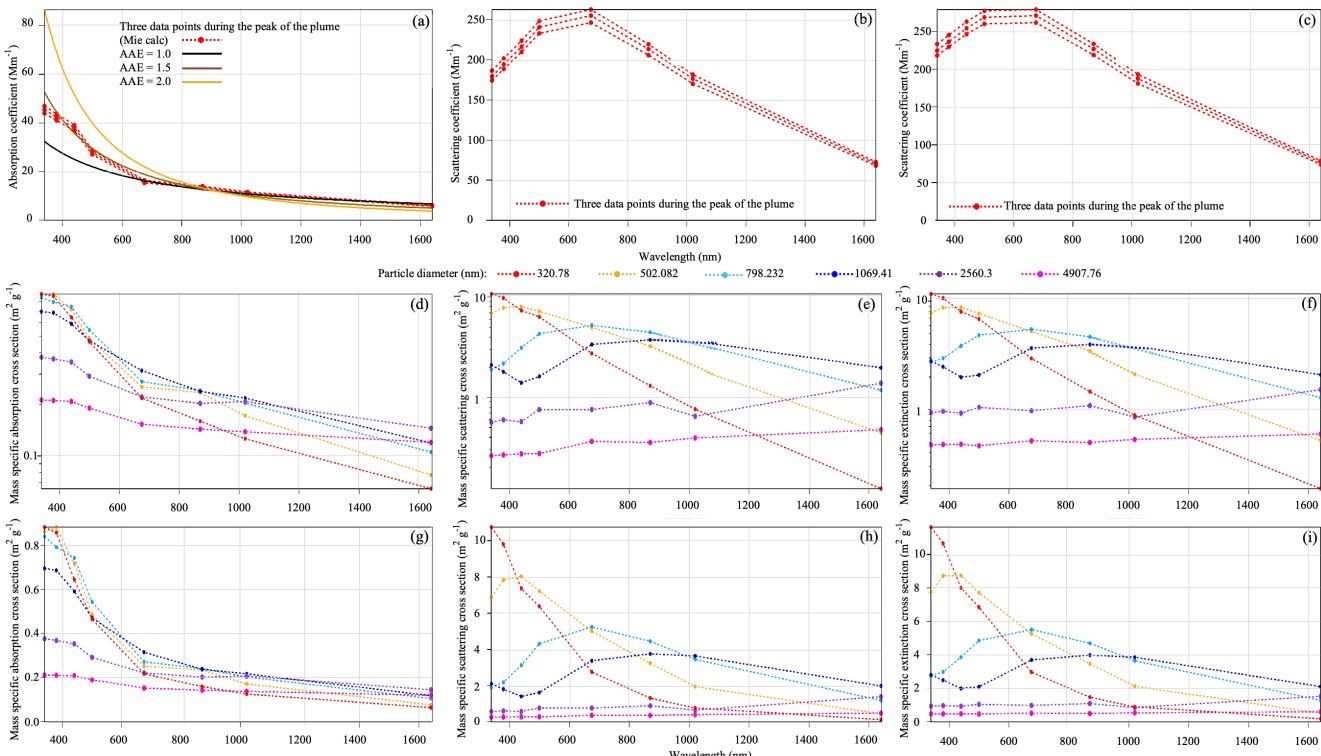

**Figure 5: Sensitivity test for the (a) extinction with contribution from (b) scattering and (c) absorption at JFJ for monodisperse assumption and (b, d, f) are the corresponding coefficient with polydisperse assumption.**

In Figure 5 a, c and e, we present approximately calculated spectral mass specific extinction, scattering, and absorption cross

sections, respectively, for perfectly monodisperse homogeneous spheres with the same complex refractive index as above, to further assess the role of material properties and particle size. Figure 5e corroborates that light absorption has a typical spectral behaviour with a negative gradient for all particle sizes. By contrast, the spectral dependence of light scattering depends strongly on particle size. The calculated MSC has a maximum in the visible for the examples with diameters between 300 nm and 800 nm, most prominently and most similarly to the observed spectral AOD for the 500 nm example. By contrast, MSC

exhibits a strong negative gradient for smaller diameters and a weak positive gradient for larger diameters. So, the observed unique spectral AOD observed in the wildfire plume is for the most part a result a rather unusual model diameter of the volume





size distribution, which exactly falls to the size range, where light scattering shows this feature. Here it is important to stress that the real part of the refractive index was chosen to be independent of wavelength, which is in agreement with the AERONET retrieval results. Light scattering also depends on the imaginary part of the refractive index, which has considerable spectral

dependence based on the AERONET retrieval result, in agreement with AAE significantly larger than unity. Therefore, we performed additional Mie calculations with the imaginary refractive index set to a constant value or to zero. The results, which are not shown here, demonstrate, that the spectral dependence of MSC and MEC is largely insensitive to these changes in the imaginary refractive index. So, we can conclude that the peak of MSC at a wavelength of around 600 nm to 700 nm for 800 nm particles is a mere particle size effect, i.e. a result of enhanced scattering near the centre of the Mie regime, whereas the

drop towards shorter wavelength or longer wavelength is caused by the transitions towards the geometric or Rayleigh regimes, respectively.

### 3.2 Tracing the smoke plume using ground measurements

### 3.2.1 High aerosol loading and unique spectral AOD curvature

We further investigated the track of the plume using ground based AOD measurements by AERONET. The description of the

AERONET stations considered in this analysis is presented in Table 1. These stations were selected on the basis of high AOD days within the considered time period, i.e., from 20 September 2023 to 05 October 2023, and on AOD peak occurring at wavelengths other than the shortest AERONET AOD measurement wavelength (i.e., 340 nm). These stations are categorised on the basis of their location, either east or west of the North Atlantic Ocean. We define smoke age based on the transport time of the plumes, i.e., the time taken by the plume to reach a location from the source, namely Canada (first visibility on 22-23

September 2023 from satellite images moving towards Eastern direction as can be observed from Fig. A2 in the Appendix).

**Table 1: Description of the AERONET stations considered in this analysis for tracing the smoke plume**

| AERONET Station name | City, Country | Code | Station Latitude (°) | Station Longitude (°) | Station Altitude (m) | Day of peak event | Smoke age (days) |
|---|---|---|---|---|---|---|---|
| Stations west of North Atlantic Ocean | | | | | | | |
| Narsarsuaq | Narsarsuaq, Denmark | NSQ | 61.16 | -45.44 | 75.0 | 25/09/2023 | 2-3 |
| Chapais | Chapais, Canada | CPS | 49.82 | -74.97 | 373.4 | 26/09/2023 | 5-6 |
| Halifax | Halifax, Canada | HFX | 44.64 | -63.59 | 65.0 | 27/09/2023 | 4-5 |
| CARTEL | Sherbrooke, Canada | CRL | 45.38 | -71.93 | 251.0 | 28/09/2023 | 3-4 |
| NEON_Bartlett | North Conway, USA | BRT | 44.06 | -71.23 | 273.0 | 28/09/2023 | 5-6 |
| Stations east of North Atlantic Ocean | | | | | | | |
| Chilbolton | Chilbolton, UK | CBT | 51.14 | -1.44 | 88.0 | 26/09/2023 | 3-4 |
| Oslo_MET_Norway | Oslo, Norway | OSL | 59.94 | 10.72 | 50.0 | 26/09/2023 | 3-4 |
| Paris | Paris, France | PAR | 48.85 | 2.36 | 50.0 | 29/09/2023 | 6-7 |
| Palaiseau | Palaiseau, France | PLS | 48.71 | 2.21 | 156.0 | 29/09/2023 | 6-7 |
| REIMS_GSMA | Reims, Paris | RGS | 49.24 | 4.07 | 133.0 | 30/09/2023 | 7-8 |
| Lille | Lille, France | LIL | 50.61 | 3.14 | 60.0 | 30/09/2023 | 7-8 |





| Payerne | Payerne, Switzerland | PAY | 46.81 | 6.94 | 491.0 | 01/10/2023 | 8-9 |
|---|---|---|---|---|---|---|---|
| Davos | Davos, Switzerland | DAV | 46.81 | 9.84 | 1589.0 | 01/10/2023 | 8-9 |
| Innsbruck_MUI | Innsbruck, Austria | INN | 47.26 | 11.39 | 620.0 | 01/10/2023 | 8-9 |
| HohenpeissenbergDWD | Hohenpeißenberg, Germany | HPB | 47.80 | 11.01 | 989.7 | 01/10/2023 | 8-9 |
| Kanzelhohe_Obs | Kanzelhoehe, Austria | KZO | 46.68 | 13.90 | 1526.0 | 01/01/2023 | 8-9 |
| Hel_IGF | Hel, Poland | HGF | 54.60 | 18.80 | 20.0 | 01/10/2023 | 8-9 |
| Moldova | Kishinev, Moldova | MDV | 47.00 | 28.82 | 205.0 | 03/10/2023 | 10-11 |

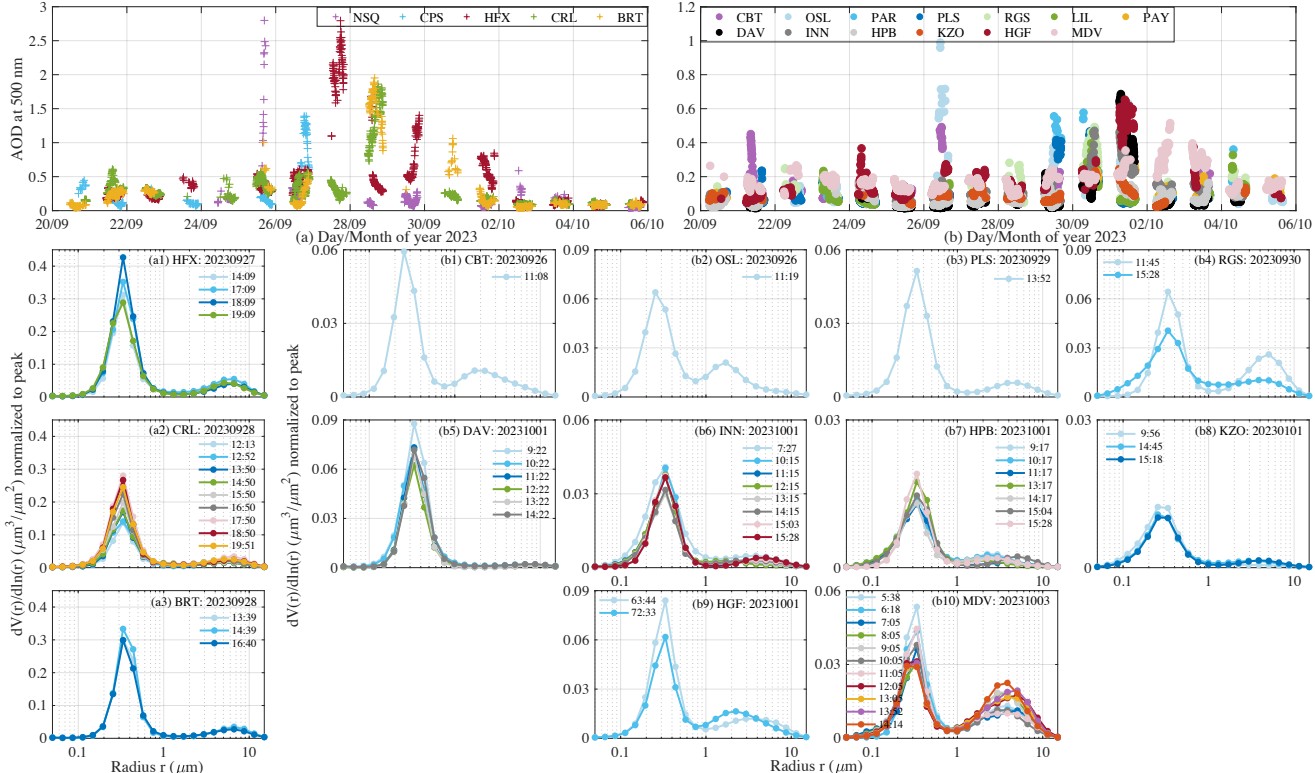

**Figure 6: Variation of AOD at 500 nm between 20 September to 05 October in 2023 at stations (a) west and (b) east of North Atlantic Ocean and the corresponding volume size distribution (VSD) as (a1-a3) and (b1-b9), respectively at different time instances (when sky error < 5%) for the day with peak AOD. The missing stations are due to either data unavailability or availability with sky error > 5% and solar zenith angle > 45 degree.**

Figure 6a and 6b present the variation of AOD at 500 nm from 20 September to 05 October 2023, at AERONET stations located west and east of the North Atlantic Ocean, respectively, as described in Table 1. The peak day of the event for each station was selected based on the observed elevated AOD values. The corresponding size distributions for each station are presented in Fig. 6 a1-b10 (the missing stations here are due to unavailability of VSD data of sufficient quality for reliable interpretation). All stations showed a greatly shifted peak in the VSD in fine mode towards larger particle size. Several studies (e.g., Zheng et al., 2020, June et al., 2022, Lu et al., 2025) have reported high fraction of particles with large diameter within accumulation size ranges, approximately ranging between 100 to 250 nm for wildfire smoke plume transports. However, the





wildfire transport case presented here showed a shifted peak in VSD in fine mode till diameters close to approximately 800
nm. Moreover, the VSD in coarse mode with almost flat curve and close to zero was observed at DAV indicating the case of
very low or no background aerosols during the plume passage at this station (properties before, during and after the peak day
of the plume is presented in Table 2).

Figure 7 shows the spectral variation of AOD on the peak day of the event at all the stations shown in Fig. 6, with an indication
of the peak AOD value at one time instance. Unlike the usual spectral AOD variation that monotonically decreases with
wavelength, the cases presented here showed a spectral curvature in AOD with peak AOD values occurring at wavelengths
other than the lowest AERONET measurement wavelength i.e., 340 nm. The noticeable stations are DAV, HFX and BRT,
with a peak in AOD at 500 nm, while other stations showed this special characteristic of spectral AOD peak at 500 nm. DAV
and BRT had a very well-defined peak at 500 nm throughout the day.

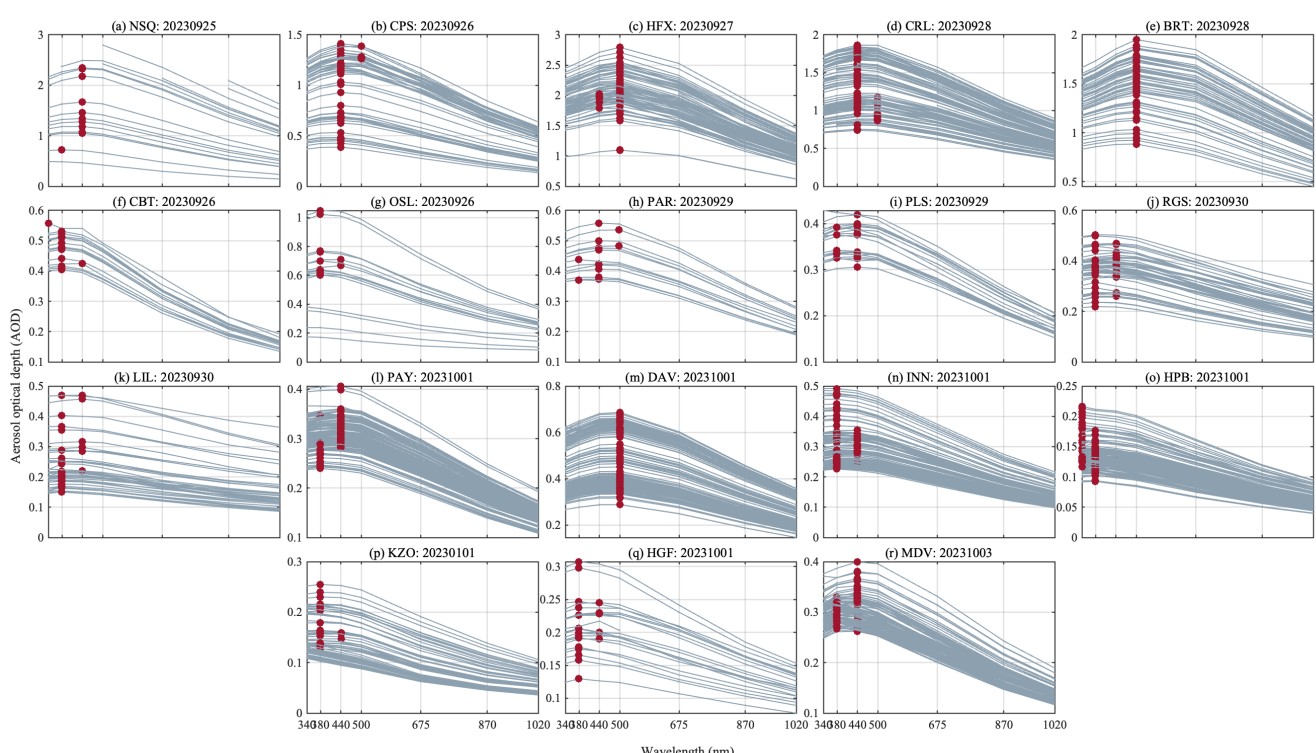


**Figure 7: (a-r) Spectral variation (curvature) of AOD on the peak day of the event at the respective stations (blue lines). Red dots indicate the peak AOD value at one time instance.**

### 3.2.2 Effect of special spectral AOD curvature on single scattering albedo

Figure 8 presents the spectral SSA variation during the peak day of the event at the respective AERONET stations and at JFJ
from in situ measurements. The spectral SSA was high (mostly above 0.9) at all wavelengths, increasing in UV-Vis region
from 440 to 675 nm, and then remaining either constant or decreasing at longer wavelengths. For dust particles, the SSA
typically monotonically increases with wavelengths (Li et. al., 2015; Collaud Coen et al., 2004; Yus-Díez et al., 2021;



Kaskaoutis et al., 2021) whereas for black carbon dominated absorbing aerosols, the SSA usually monotonically decreases with wavelength (Li et. al., 2015; Chauvigné et al., 2019; Kaskaoutis et al., 2021). However, nonmonotonic spectral curvature in SSA is dominant in East Asia, frequently peaking at 675 nm, as reported by Li et. al. (2015) and attributed to enhanced absorption by some dust aerosol species in the UV region that become non-absorbing in the visible part of the spectrum and black carbon dominant absorption at longer wavelengths. The uniqueness of the case in the present manuscript is that similar nonmonotonic spectral SSA curvature occurred under conditions of very low or absent coarse mode size distributions (as presented in Fig. 4 and Fig. 6), i.e., the SSA spectral curvature was associated with large fine mode particles.

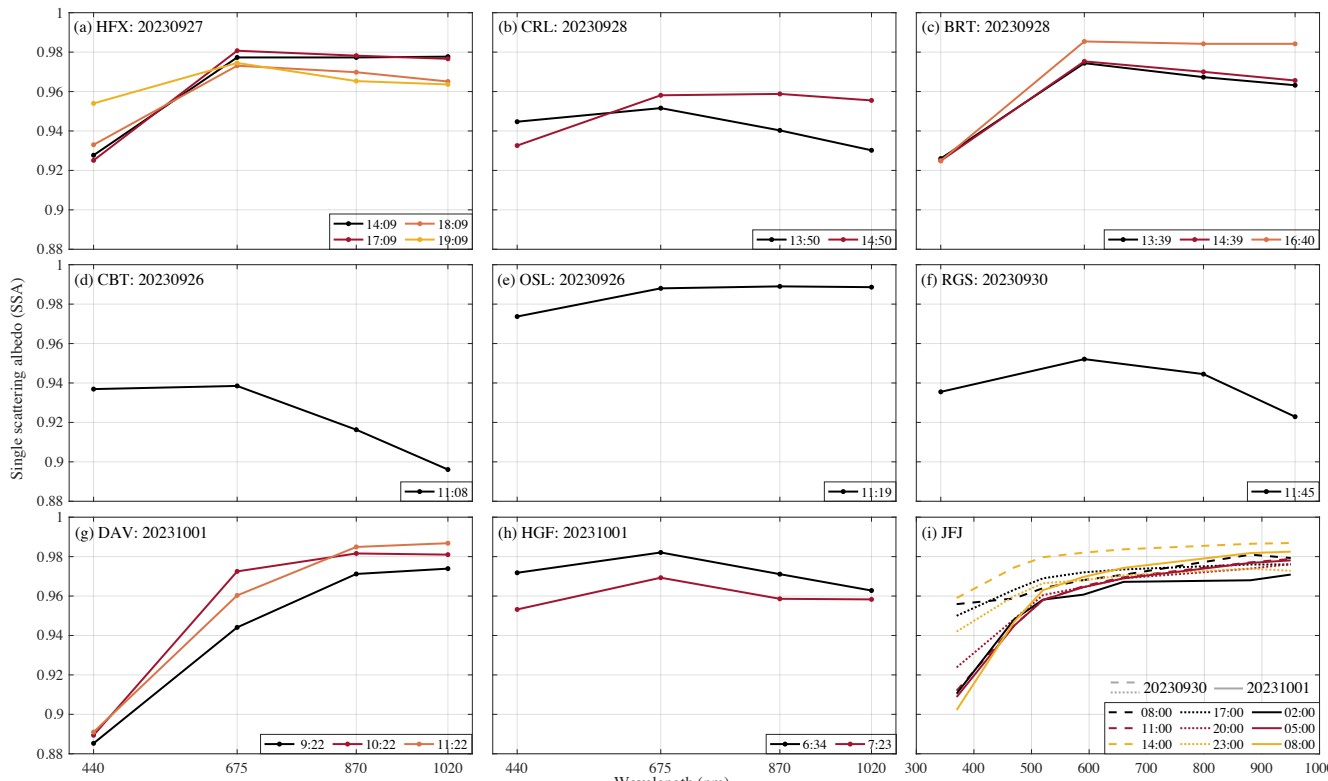

**Figure 8: (a-h) Variation of SSA during the peak day of the event at the respective stations (only when sky error < 5% and coincident AOD at 440 nm > 0.4) from AERONET and (i) at JFJ from in-situ measurement. The missing stations are due to either unavailability of SSA on that day or SSA available with sky error > 5% and AOD at 440 nm < 0.4 in case of AERONET.**

As presented by the authors in Giles et al. (2012), fine mode particles containing hygroscopic aerosol particles such as sulfates have a nearly neutral SSA spectral dependence (Dubovik et al., 2002), while those containing black carbon aerosol particles as the sole absorber exhibit a decreasing SSA with wavelength, and those composed of brown carbon (BrC) or organic carbon (OC) aerosol particles as sole absorber exhibit an increasing SSA with wavelength (strong absorption in ultraviolet and visible region) (Eck et al., 2009). Moreover, a varying concentration of BC with dust, BrC, and/or OC aerosol particles can lead to ambiguous (increasing, decreasing, or constant) SSA dependence on wavelength depending on the spectral absorption



properties of the aerosol mixture with the net effect being stronger absorption across the retrieved spectrum (e.g., 440 to 1020
        nm) (Dubovik et al., 2002; Giles et al., 2011; Giles et al., 2012).

        In the case presented here, the SSA concave spectral curvature for smoke aerosol presents a similar spectral behaviour as
        observed for absorbing dust aerosols in the UV-Vis region below 675 nm. In the case of dust aerosols, pure coarse mode dust
        usually shows an increasing SSA with wavelength whereas a mixing of dust with BC or other pollution leads to the SSA

concave curvature as presented by Li et al. (2015); however, this concave spectral curvature in SSA (due to absorbing
        characteristics of coarse mode particles) did not produce spectral curvature in AOD. Another study over the Himalayan region
        by Tian et al. (2023) also observed a spectral curvature in SSA and scattering coefficient but the absorption coefficient
        monotonically decreased with wavelength for an aerosol size distribution indicating the presence of both fine and coarse mode
        particles during the studied biomass burning event. However, in the present manuscript, most of the stations (Fig. 8) showed

an SSA concave spectral curvature except DAV and JFJ, which show an SSA monotonically increasing with wavelength (a
        spectral behaviour similar to pure dust case), and these are associated with the concave spectral curvature in AOD for fine
        mode aerosols as presented in Fig. 6 and Fig. 7.

        The case of DAV (from remote sensing measurements) and JFJ (from in situ measurements) as presented in Fig. 8g and Fig.
        8i showed a dust-aerosol-like SSA spectral variation, with SSA monotonically increasing with wavelength, for large fine mode

smoke aerosols in the absence of coarse mode aerosols (Fig. 6b5). This suggests that in this particular transported plume, the
        peculiar size distribution with the large fine mode aerosols lead to the peculiar increasing SSA with wavelength.

### 3.2.3 Explanation for spectral AOD peak at 500 nm: Fresh/slightly aged vs aged smoke

        In this section, we discuss the stations that displayed a peak in AOD at 500 nm, categorizing the transported smoke plume as
        fresh or slightly aged in case of the stations near the fire source i.e., HFX and BRT, and aged smoke in case of the station

located far away from the fire source i.e., DAV.

        HFX is a coastal station, BRT is characterized by the presence of hills and mountains in the area and DAV is a high mountain
        location. The size distribution at DAV was dominated by the fine mode with no discernible coarse mode, while HFX and BRT
        had a large fine mode and a very small coarse mode (see Fig. 6) during the studied BB event.

        Table 2 presents the observed aerosol properties for fresh/slightly aged (at HFX and BRT) and aged smoke (at DAV) plume

during peak day of the event. HFX and BRT were characterized by extremely high AOD values during the peak day of the
        event with almost not much difference (less than 0.01) in negative AOD AE values in wavelength pair of 340-440 nm and
        380-500 nm and a non-monotonic SSA concave spectral curvature with peak at 675 nm. On the contrary, DAV had high AOD
        (with respect to the regional climatological values) but not of the magnitude as HFX and BRT with approximately 0.02
        difference in negative AE for the wavelength pair of 340-440 nm and 380-500 nm and a monotonically increasing SSA with

wavelength (which is a typical characteristic of coarse mode aerosol particles).



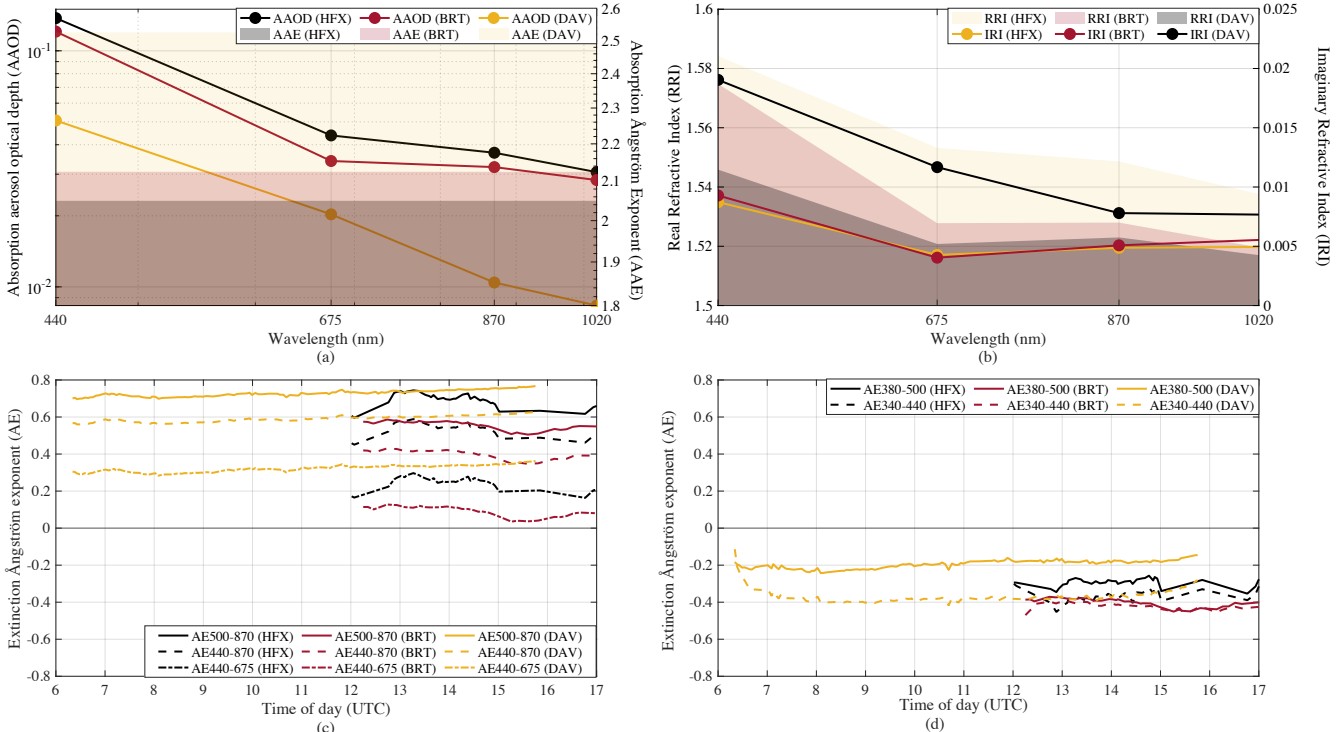

**Figure 9: Average (a) absorption AOD (shown as line) and absorption AE (shown as area) in logarithmic axis and (b) real (shown as area) and imaginary (shown as line) refractive indices and the variation of extinction AE in the spectral range that exhibited (c) positive and (d) negative values at stations HFX, BRT and DAV during the peak day of the event based on data availability from AERONET inversion products.**

Figure 9 shows the absorption AOD and absorption AE averages during the peak day of the event for these three stations discussed in this section, as well as the real and imaginary parts of the refractive indices. For fresh/slightly aged smoke at HFX and BRT, the absorption AOD at 440 nm was approximately 1.4, with absorption AE (AAE) in the wavelength range 440-870 nm being close to 2.0. In contrast, for aged smoke at DAV, the corresponding AAOD and AAE were observed to be lower (close to 0.05) and higher (close to 2.5), respectively. There is steeper AAOD variation with wavelength on logarithmic scale that corresponds to higher AAE values. The AAE for wavelength range 370–880 nm at JFJ as observed from in situ measurements (presented in Section 3.1.3) was approximately 2 which is in good agreement with the remote sensing measurement based AAE presented here, considering the uncertainty and difference in the wavelength ranges associated with these two types of retrievals. The imaginary part of the refractive index values was almost identical for HFX and BRT (approximately 0.005) being mostly flat from 1020 to 675 nm, and slightly increased to approximately 0.009 from 675 to 440 nm. In contrast, at DAV it was higher, increasing from 0.008 to 0.019 from 1020 to 440 nm. This could also be indicative of the presence of BrC, as the imaginary part of the refractive index of BrC change with aging and coating. The real part of the refractive index was highest at HFX (approximately 1.58 at 440 nm), followed by BRT (approximately 1.57 at 440 nm), and



lowest at DAV (approximately 1.55 at 440 nm) for the wavelength range between 440-1020 nm. Hence, it indicates the presence of highly absorbing aerosols near the source for fresh/slightly aged smoke plume. However, the long range transported smoke plume had lower absorption AOD (AAOD) and higher contribution of AAOD to AOD at 440 nm (Table 2). The imaginary part of the refractive index (IRI) decreases with wavelength, which indicates the presence of BrC. This is consistent with the AAE from in situ observation being substantially larger than unity as presented in Section 3.1.3 (BC with constant IRI has an AAE of around unity). Also, due to the very high SSA at longer wavelength, indicating a very low BC fraction, the

SSA also remains high at shorter wavelength despite BrC contributing to absorption.

Figure 9 (c, d) shows the variation of extinction AE with different wavelength pairs at 440-870 nm, 380-500 nm, 440-675 nm, 500-870 nm and 340-440 nm during the peak day of the event at HFX, BRT and DAV. The AE at all wavelength pairs was observed to be much lower than for typical smoke aerosols. The highlight here is the negative AE in wavelengths pairs involving 340 and 380 nm i.e., wavelengths corresponding to the UV part of the electromagnetic spectrum, and low AE with

longer wavelength pairs (also shown in Fig. A3 in Appendix for all stations). This negative AE is associated with the concave spectral curvature in AOD following the logarithmic spectral variation of AOD using the Ångström power law [Ångström 1929; Martínez-Lozano et. al., 1998]. The change in sign of extinction AE for the wavelength ranges shown in Figure 9c and 9d is a result of the maxima in AOD at around 500 nm, which is driven by the large particle diameter of the dominant fine mode aerosol.

Another point to note in this smoke plume transport is that the fire plume was visible in the MODIS true colour images (Fig. A2 in Appendix) even in the presence of clouds which indicates that the injection height of this Canadian wildland fire was quite high. Zhang et. al. (2024) confirmed high pyrocumulonimbus (pyroCb) activity (a cumulonimbus clouds formed by wildfires-driven convection) activity during the Canadian wildfire season of 2023; however, only a minor amount of aerosol managed to reach above the troposphere, with most convective events limited to the upper troposphere. Black carbon (BC) is

one of the main constituents associated with pyroCb-injected smoke aerosols. A study by Beeler et. al. (2024) revealed that a distinguishing feature of BC-containing particles associated with pyroCb clouds can develop thick external coatings of >200 nm thick (made of condensed organic matter) and also other studies showed that BC particles coating thickness can vary between <30 nm to 400 nm (Katich et. al., 2023). Beeler et. al. (2024) also reported that pyroCb BC (approximately 18.3) has large average ratio of coating mass to BC mass as compared to the urban (approximately 1.8) and usual wildfire source-based

BC (approximately 7.3). A study by Huang et. al. (2024) found that atmospheric aging leads to greater heterogeneity in coating-to-BC mass ratios, reshaping particles to be more spherical and compact, and the heterogeneity in size, coating and non-spherical shapes may account for about 20% and 30% of the observed reduction in BC absorption capacity, respectively. Another study by Dahlkötter et al. (2014) revealed a quite concentrated transatlantic plume probe over Europe, for which BC was shown to be heavily coated, which indicates photo-chemical production and condensation of SOA along the transport.

Similar special spectral AOD variation was observed in another wildfire event in California/Oregon in 2020 as presented by Eck et al., 2023 where the authors described the strong presence of coated black carbon and/or BrC in these transported plumes that stayed over the Pacific Ocean for some time before turning back to the land surface. HFX and BRT stations of the present





manuscript seem to be similar to the Eck et. al., 2023 case, while DAV is very special concerning the monotonically increasing SSA with wavelength despite no visible signatures of coarse mode particles.

For selective absorbers like BrC, which typically absorb in the UV-Vis region and almost non absorbing in the near infrared part of the spectrum, the absorption typically decreases with wavelength. Therefore, the observed special AOD spectra can be related to the dominant aerosol mode with an unusually large diameter at the upper end of the fine mode. But the question is how did the size distribution of aerosol particles reach this extreme size distribution during this particular long range transported wildfire smoke, with non present or very small visible coarse mode? A central feature is very low plume dilution,

very high aerosol and likely also VOC mass concentrations over multiple days. This enables sustained growth by condensation and/or coagulation.

**Table 2: Aerosol property for fresh/slightly aged and aged smoke plume during peak day of the event as mean (in brackets are the standard deviation). These properties are for before, during and after the peak day of rare smoke plume transport i.e., special concave spectral curvature in AOD peaking at 500 nm.**

| Property/wavelength | Fresh/slightly aged smoke | | | | | | Aged smoke | | |
| --- | --- | --- | --- | --- | --- | --- | --- | --- | --- |
| | HFX | | | BRT | | | DAV | | |
| | Before | During | After | Before | During | After | Before | During | After |
| Aerosol optical depth (AOD) | | | | | | | | | |
| 340 | - | 1.86(0.28) | 0.52(0.29) | - | 1.33(0.20) | 0.37(0.00) | 0.15(0.04) | 0.44(0.11) | 0.04(0.01) |
| 380 | 0.52(0.10) | 1.94(0.29) | 0.51(0.31) | - | 1.38(0.21) | 0.36(0.00) | 0.15(0.04) | 0.46(0.11) | 0.04(0.01) |
| 440 | 0.51(0.09) | 2.04(0.31) | 0.49(0.33) | - | 1.47(0.24) | 0.34(0.00) | 0.14(0.04) | 0.48(0.12) | 0.04(0.01) |
| 500 | 0.48(0.09) | 2.08(0.32) | 0.46(0.34) | - | 1.53(0.26) | 0.31(0.00) | 0.13(0.04) | 0.48(0.12) | 0.04(0.01) |
| 675 | 0.36(0.07) | 1.85(0.28) | 0.34(0.31) | - | 1.41(0.27) | 0.22(0.00) | 0.10(0.03) | 0.43(0.11) | 0.03(0.01) |
| 870 | 0.24(0.05) | 1.40(0.22) | 0.23(0.24) | - | 1.10(0.22) | 0.15(0.00) | 0.08(0.03) | 0.32(0.08) | 0.02(0.01) |
| 1020 | 0.18(0.04) | 1.08(0.17) | 0.17(0.19) | - | 0.87(0.18) | 0.12(0.00) | 0.07(0.02) | 0.25(0.07) | 0.02(0.01) |
| Extinction Ångström exponent (AE) | | | | | | | | | |
| 440-870 | 1.09(0.12) | 0.59(0.15) | 1.27(0.19) | - | 0.44(0.07) | 1.18(0.00) | 0.84(0.23) | 0.59(0.02) | 0.74(0.13) |
| 380-500 | 0.30(0.14) | -0.26(0.07) | 0.51(0.20) | - | -0.35(0.09) | 0.55(0.00) | 0.62(0.27) | -0.19(0.02) | 0.53(0.18) |
| 440-675 | 0.87(0.13) | 0.28(0.08) | 1.08(0.22) | - | 0.14(0.08) | 1.02(0.00) | 0.81(0.25) | 0.32(0.02) | 0.76(0.13) |
| 500-870 | 1.23(0.12) | 0.74(0.12) | 1.38(0.19) | - | 0.59(0.07) | 1.27(0.00) | 0.85(0.23) | 0.73(0.02) | 0.72(0.13) |
| 340-440 | 0.14(0.15) | -0.35(0.05) | 0.27(0.16) | - | -0.38(0.07) | 0.30(0.00) | 0.45(0.24) | -0.37(0.04) | -0.04(0.16) |
| Single scattering albedo (SSA) | | | | | | | | | |
| 440 | 0.92(0.02) | 0.93(0.01) | 0.91(0.01) | - | 0.92(0.00) | - | - | 0.89(0.00) | - |
| 675 | 0.93(0.03) | 0.98(0.00) | 0.92(0.01) | - | 0.98(0.00) | - | - | 0.96(0.01) | - |
| 870 | 0.91(0.03) | 0.97(0.00) | 0.89(0.02) | - | 0.97(0.01) | - | - | 0.98(0.01) | - |
| 1020 | 0.90(0.04) | 0.97(0.01) | 0.88(0.02) | - | 0.97(0.01) | - | - | 0.98(0.00) | - |
| Absorption aerosol optical depth (AAOD) | | | | | | | | | |
| 440 | 0.04(0.01) | 0.14(0.03) | 0.05(0.03) | - | 0.12(0.00) | - | - | 0.05(0.01) | 0.01(0.00) |
| 675 | 0.03(0.01) | 0.04(0.01) | 0.03(0.03) | - | 0.03(0.01) | - | - | 0.02(0.01) | 0.00(0.00) |
| 870 | 0.02(0.01) | 0.04(0.01) | 0.03(0.03) | - | 0.03(0.01) | - | - | 0.01(0.01) | 0.00(0.00) |
| 1020 | 0.02(0.01) | 0.03(0.01) | 0.02(0.02) | - | 0.03(0.01) | - | - | 0.01(0.01) | 0.00(0.00) |
| Absorption Ångström exponent (AAE) | | | | | | | | | |
| 440-1020 | 1.26(0.86) | 2.05(0.68) | 1.08(0.34) | - | 2.12(0.49) | - | - | 2.52(0.86) | 1.24(0.44) |
| X=AAOD/AOD% | | | | | | | | | |
| 440 | 7.84 | 6.86 | 10.20 | - | 8.16 | - | - | 10.42 | 25.00 |
| 675 | 8.33 | 2.16 | 8.82 | - | 2.13 | - | - | 4.65 | 0.00 |





| | | | | | | | | | |
|---|---|---|---|---|---|---|---|---|---|
| 870 | 8.33 | 2.86 | 13.04 | - | 2.73 | - | - | 3.12 | 0.00 |
| 1020 | 11.11 | 2.78 | 11.76 | - | 3.45 | - | - | 4.00 | 0.00 |
| Volume concentration ($\mu m^3/\mu m^2$) | | | | | | | | | |
| Fine | 0.06 | 0.24 | 0.05 | - | 0.19 | - | 0.01 | 0.07 | 0.00 |
| Coarse | 0.01 | 0.06 | 0.01 | - | 0.04 | - | 0.01 | 0.00 | 0.00 |
| Total | 0.07 | 0.30 | 0.07 | - | 0.23 | - | 0.02 | 0.07 | 0.01 |
| Effective radius ($\mu m$) | | | | | | | | | |
| Fine | 0.24 | 0.30 | 0.22 | - | 0.33 | - | 0.22 | 0.31 | 0.27 |
| Coarse | 3.34 | 3.60 | 2.75 | - | 3.78 | - | 2.79 | 3.30 | 3.46 |
| Total | 0.29 | 0.37 | 0.27 | - | 0.39 | - | 0.39 | 0.33 | 0.41 |
| Volume median radius ($\mu m$) | | | | | | | | | |
| Fine | 0.26 | 0.33 | 0.24 | - | 0.35 | - | 0.27 | 0.34 | 0.30 |
| Coarse | 4.05 | 4.51 | 3.47 | - | 4.74 | - | 3.34 | 4.68 | 3.97 |
| Total | 0.43 | 0.55 | 0.41 | - | 0.55 | - | 0.88 | 0.39 | 0.82 |


Such extreme size distributions have been observed in some of the past volcanic events, which led to the appearance of blue/green sun. A study by Wullenweber et. al. (2021) and references therein described that narrow aerosol particle size distributions narrow, centred around a radius of 500 nm, can cause anomalous scattering (i.e., increase in scattering cross sections with wavelength in the visible part of the spectrum). This work was associated with the appearance of blue coloured

sun post volcanic eruption (e.g., Krakatao in 1883) or massive forest fires due to Rayleigh scattering as well as minor impacts of water vapour and ozone. Even though the phenomenon of occurrence of blue sun was not observed in the Canadian plume transport analysed in the present manuscript at the extremely special features presented at DAV on 01 October 2023, this was probably the closest extreme condition reached for this phenomenon to occur in the history of measurement of DAV. Wullenweber et. al. (2021) observed that a low refractive index combined with a larger particle radius range led to stronger

anomalous scattering and increased the probability of occurrence of blue sun phenomenon. The observed real refractive index in the case of occurrence of blue sun phenomenon as reported by some previous studies by Penndorf (1953) and Ball et al. (2015) was around 1.46 and 1.50 and hence considered to be in the range of 1.3 to 1.5. However, in the case of DAV, the real refractive index was above 1.5 at all wavelengths that might explain the non-occurrence of blue sun phenomenon in this Canadian smoke plume transport.

As reported by Wullenweber et. al. (2021), unfortunately no aerosol samples are available from the past occurring extreme size distribution events, therefore the true aerosol optical properties associated with such events are unknown. Hence, the aerosol optical properties observed during the Fall 2023 Canadian wildfire smoke plume transport as presented in this manuscript as well as the California/Oregon wildfires smoke plume transport in 2020 as presented by Eck et. al. (2023) can serve as a probable sample for future studies exploring this unusually extreme size distribution occurrence and associated

anomalies.




### 3.3 Satellite observation and model evaluation of the event

In this section, we analyse how well the satellite and model reanalysis captured this special smoke plume transport event. Figure 10 presents a comparison between the AERONET AOD and satellite based AOD from MODIS and model reanalysis based AOD from MERRA2 at 550 nm for all stations during the study period from 20 September to 05 October 2023, with

AERONET AOD at 500 nm being interpolated to 550 nm using Ångström power law for comparability (the statistics are also presented in Table A3 in Appendix). For comparison of MODIS and MERRA2 AODs with AERONET AOD, the daily mean values were considered from AERONET.

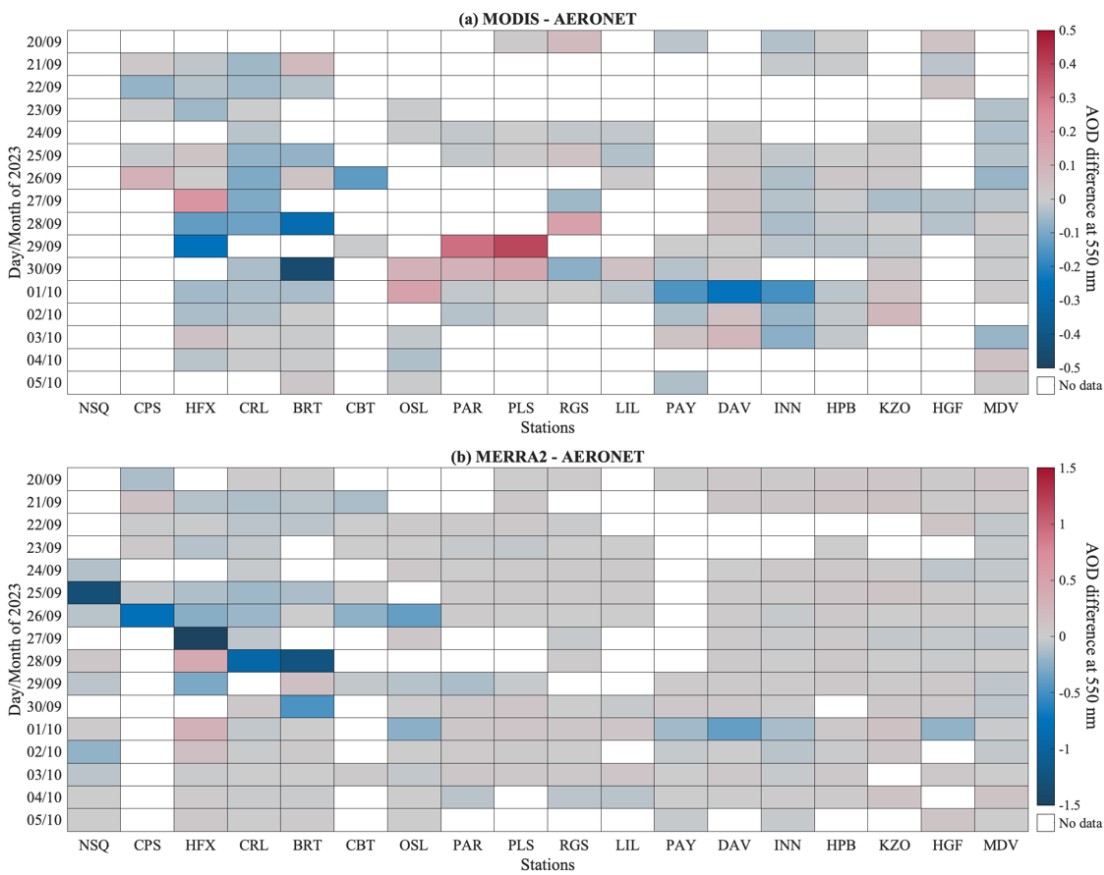

**Figure 10: Daily mean AOD difference of AERONET AOD from (a) MODIS and (b) MERRA2 AOD at 550 nm for all stations from**
**20 September to 05 October 2023. AERONET AOD at 500 nm is interpolated to 550 nm using Ångström power law for comparability.**

From MODIS (MODIS AOD at 550 nm is shown in Fig. A4 in Appendix) (as compared to AERONET measured AOD), it was found to have overestimation of AOD for 6 stations and underestimation for the remaining 10 whose data were available during the peak day of the event at respective stations. MODIS retrieval algorithm seems to capture the AOD with lesser error

at stations near the fire source with absolute and relative percentage error below approximately 0.3 and 20%, respectively. While the stations with long range transported smoke plume had relatively higher error with overestimation reaching up to



approximately 0.4 and underestimation up to 0.3. Another observation is that INN and HPB lie on the same pixel in MODIS retrieval with underestimation of 0.17 (approximately 63%) and 0.02 (approximately 17%), respectively (these two locations are mountainous, but the plume properties were not identical), and also PAR and PLS with overestimation of 0.31

(approximately 74%) and 0.39 (approximately 118%), respectively. This highlights the fact that one pixel of satellite resolution can have diversity and it can be quite challenging to retrieve parameters accurately specially during extreme and rare events. Some causes of the underestimation or overestimation of AOD from real situation (AERONET measurements) by MODIS could be due to inhomogeneity within a pixel, ground reflectance, high altitude or presence of high absorption layer e.g., in this case was thick smoke plume.

For MERRA2, in most of the cases, there was underestimation of AOD as compared to AERONET measured AOD with the highest differences observed for the station near the fire source in Northern Canada varying from approximately 0.8 to 1.5 (CPS, CRL, BRT, NSQ and HFX, respectively). While the stations towards the east of the North Atlantic Ocean characterised by long range smoke aerosol transport had the underestimation of MERRA2 AOD below approximately 0.4 and three stations showed overestimation but below approximately 0.1 (Ceamanos et. al., 2023; Shang et. al., 2024).

AERONET has been used in various studies to evaluate MODIS AOD data and MODIS uncertainty is defined based on this comparison (Sayer et al., 2013 and references there in). However, based on the results of Sayer et al. (2013), there are differences due to various reasons, including spatial comparison aspects, retrieval aspects etc. In addition, the vast majority of these data represent cases with relatively low AODs. In addition, good evaluation results of a satellite-based instrument based on ground-truth, does not necessarily mean agreement in a very specific and rare case as the one presented in this manuscript.

It is evident from Fig. 10 that for various stations the agreement of MODIS and AERONET AOD at the days affected by Canadian smoke differ from other days. This analysis suggests that in special cases, the defined uncertainties of an instrument based on statistical differences with another may vary. MODIS and MERRA2 are interconnected in a way that the MERRA2 involves assimilation from MODIS as well as other observations. However, there still can be differences in the AOD estimations from the two algorithms in such rare situations as is also evident that the two panels of Fig. 10 differ.

This analysis illustrates the importance and need of experimental and ground-based measurements to capture such rare extreme weather events during which there can be significant underestimation and overestimation from satellite and model reanalysis datasets.

## 4 Conclusions

The more frequent heatwaves and intensified drought seasons due to the ongoing climate change have led to widespread and
more intense wildfires, transforming forests into natural carbon sources rather than designated carbon sinks. The 2023 Canadian wildfires were extreme and unique in scale and intensity, primarily driven by widespread hot and dry weather, with 2023 recorded as the driest and warmest year since at least 1980. Long range transport of wildfire smoke to Europe from North America is not a rare event anymore. However, the Canadian wildfires of Fall 2023 displayed the transport of a highly



concentrated plume to Europe along with observation of special spectral aerosol optical depth (AOD) characteristics and other aerosol properties.

The 2023 Canadian wildfire smoke plume reached the Swiss Alps between late September and the beginning of October after crossing the North Atlantic Ocean. The plume has been detected by the ground-based remote sensing instrumentation installed at the PAY and DAV stations, i.e., by the Raman lidar RALMO and the ceilometer, respectively. The 20-year climatological spectral AOD variation at DAV showed mean AOD values below 0.11 and only approximately 1% of the total AOD

measurements above 0.5. The daily mean spectral AOD values at DAV showed peak in AOD at 500 nm only for one day on 01 October 2023. The mass size distribution at DAV on 01 October 2023 and JFJ on 30 September 2023 showed a strong shift of the peak towards larger particle diameters within the fine mode, with almost no visible coarse mode which is a very rare event. Multiwavelength light absorption measurements at JFJ showed substantial absorption at the shorter wavelengths and relatively large absorption Ångström exponent of approximately 2 for wavelength range 370 – 880 nm and scattering Ångström

exponents of approximately 0 for wavelength range 450 – 700 nm. The SSA Ångström exponents were observed to be approximately -0.10 during the plume period. High UV absorption (370 nm) than visible absorption indicated the presence of brown carbon and/or tar balls, which have a strong spectral dependence in the imaginary refractive index.

Further tracing of the smoke plume with ground-based aerosol measurements for special spectral AOD non-monotonic variation with wavelength (concave spectral curvature of AOD) revealed that the stations DAV, HFX and BRT had a peak in

AOD at 500 nm, while other stations had a peak at 440 nm or shorter wavelength longer than 340 nm, which is the minimum measurement wavelength for AERONET. Analysis of the effect of special spectral AOD concave curvature on Ångström exponent showed unusually low AE values for smoke aerosols (usually AE for smoke aerosol cases is close to or above 1.5) and even negative AE in wavelengths pairs involving 340 and 380 nm.

Another highlight of this analysis has been the observation of non-monotonic concave spectral curvature in SSA peaking at

675 nm at all the stations except DAV and JFJ where SSA monotonically increased with wavelength for fine mode aerosols which is typical characteristic of coarse mode dust aerosols. The non-monotonic concave spectral curvature in SSA peaking at 675 nm for large fine mode particle size smoke aerosols suggested the presence of Brown Carbon in the plume. The monotonically increasing SSA for fine mode particles with almost no visible coarse mode at DAV and JFJ suggested that smoke aerosol particles typical absorbing characteristic was either lost or was supressed by scattering characteristics at all

wavelengths due to large particle size while reaching DAV and JFJ.

The highlight stations of this analysis have been HFX and BRT categorized as fresh/slightly aged case and DAV as aged smoke case. The key feature of fresh/slightly aged smoke at HFX and BRT was extremely high AOD values with very little difference (less than 0.01) in negative AE values at 340-440 nm and 380-500 nm and a non-monotonic SSA concave spectral curvature peaking at 675 nm. On the other hand, the aged smoke at DAV was characterized by high AOD (with respect to the regional

climatological values), but not as high as observed at HFX and BRT with approximately 0.02 difference in negative AE at 340-440 nm and 380-500 nm and a monotonically increasing SSA with wavelength (a special behaviour for fine mode particles, that is a typical characteristic of coarse mode dust particles, as described above).



Similar concave spectral AOD curvature was observed in another wildfire event in California/Oregon in 2020 as presented by Eck et al. (2023) in which the authors suggested the presence of coated black carbon and/or BrC. For the cases presented in
this study, HFX and BRT seems to be similar to Eck et. al. (2023) case, while DAV and JFJ are quite special with respect to the observed monotonically increasing SSA with wavelength despite no visible signatures of coarse mode particles. Therefore, the observed rare AOD spectra is driven by extreme size distribution in fine mode.

Similar extreme size distributions have been observed in some of the past volcanic activities, leading to the appearance of blue/green sun, associated with anomalous scattering (i.e., increasing scattering cross sections with wavelength in the visible
part of the spectrum). The phenomenon of occurrence of blue sun was not observed in the presented Canadian plume transport at the special features in aerosol properties presented at DAV on 01 October 2023. The latter had the real refractive index above 1.5 at all wavelengths, whereas the occurrence of blue sun phenomena has been associated with refractive index below 1.5.

Future studies could investigate the catalytic conditions and mechanisms that allowed aerosol particles to reach such extreme
sizes, causing a peak in AOD at 500 nm both near the source and after long-range transport to distant location. Another question associated with the occurrence of this phenomenon could be whether there can be some relation with the Fall season as the event reported here occurred towards the end of September 2023 as well as the event of California/Oregon fires presented by Eck et al., 2023 also occurred in September of 2020. This uncertainty in understanding such rare observations highlights the importance of ground based optical and chemical aerosol measurements to better understand the impacts caused by extreme
climate events and to identify the processes (whether due to chemical species or catalytic effects during aging) that lead to extreme fine mode aerosol size distributions in wildfire plume transport.

## Abbreviations

| | |
|---|---|
| AAE | absorption Ångström exponent |
| AAOD | absorption aerosol optical depth |
| ACTRIS | Aerosol, Clouds and Trace Gases Research Infrastructure |
| AE | Ångström exponent |
| AERONET | Aerosol Robotic Network |
| AOD | aerosol optical depth |
| APS | aerodynamic particle sizer |
| BTS | BiTec Sensor |
| FWHM | full width at half maximum |
| GAW | Global Atmospheric Watch |
| GHG | greenhouse gases |
| MERRA2 | Modern-Era Retrospective Analysis for Research and Applications, Version 2 |
| MODIS | MODerate resolution Imaging Spectroradiometer |
| MPSS | mobility particle size spectrometer |
| NABEL | Swiss National Air Pollution Monitoring Network |
| PFR | Precision Filter Radiometer |
| RALMO | RAman Lidar for Meteorological Observations |
| RI | refractive indices |
| SSA | single scattering albedo |
| ToF-ACSM | Time-of-Flight Aerosol Chemical Speciation Monitor |



| VSD | volume size distribution |
| WORCC | World Optical depth Research and Calibration Center |

**Data availability:** PFR AOD data from 2004-2022 used in this analysis can be found at doi:10.48597/E5WB-NXRG,
doi:10.48597/X962-H2CJ, doi:10.48597/U3MD-VUVP, doi:10.48597/9PGM-VJZR and doi:10.48597/SXCN-DHVA. The
AERONET direct sun and inversion data used in this work are available through the portal at https://aeronet.gsfc.nasa.gov/cgi-
bin/webtool_aod_v3 (Last accessed: April 26, 2025). For the remaining data, authors can be contacted.

**Author contribution:** AM wrote the overview of the paper, and AM, SK coordinated the paper writing. AM, SK, MG, JG
conceptualized the initialization of the paper. RLM, MGB performed in situ measurements analysis and interpretation. MCC,
FNG, GM performed in situ measurements analysis and lidar measurements analysis. AM, SK, JG performed remote sensing
measurements analysis and interpretation. AM performed plume tracing analysis, satellite data and model reanalysis
assessment, and Hysplit modelling. All authors participated in writing and revision of the paper.

**Ackowledgement:** Authors would like to acknowledge the Aerosols, Clouds, and Trace gases Research Infrastructure
(ACTRIS) Switzerland project supported by the Swiss State Secretariat for Education Research and Innovation. In situ
observations at Jungfraujoch received further financial support from MeteoSwiss in the framework of the Global Atmosphere
Watch (GAW) program of the World Meteorological Organization (WMO). The International Foundation High Altitude
Research Stations Jungfraujoch and Gornergrat (HFSJG) is acknowledged for hosting the observations at the Jungfraujoch
High Altitude Research Station. SK, AM, NK would like to acknowledge HARMONIA (International network for
harmonization of atmospheric aerosol retrievals from ground-based photometers; grant no. CA21119), supported by COST
(European Cooperation in Science and Technology). AM would like to acknowledge NASA postdoctoral program fellowship
and Oak Ridge and Associated Universities. The authors would like to acknowledge the AERONET and other local instrument
operators whose data has been used in this work. The authors would like to thank Dr. Thomas Eck from NASA Goddard Space
Flight Center who provided valuable insights in understanding the concepts as presented by him in Eck et. al., 2023.



**Appendix A**

**Table A1: Summary of gas and aerosol concentrations and properties measured in situ at JFJ before, during, and after passage of the smoke plume. The average value is reported for each quantity along with the standard deviation in brackets. The before period is defined from 13:00 September 29 to 07:00 September 30; the during period from 07:00 September 30 to 08:00 October 1; and the after period from 08:00 October 1 to 03:00 October 2. The peak of the plume occurred at 09:00 on September 30.**

| Component/Property | Unit | Before (n=17 h) | Peak (n=1 h) | During excl. peak (n=22 h) | After (n=19 h) |
|---|---|---|---|---|---|
| Carbon monoxide (CO) | ppb | 90.16 (5.80) | 308.770 | 109.70 (16.76) | 70.79 (3.13) |
| Benzene | ppt | 15.37 (2.99) | | 37.36 (18.59) | 6.30 (1.42) |
| Particulate Matter (PM1) | $\mu g/m^3$ | 0.58 (0.56) | 30.95 | 1.32 (0.95) | 0.17 (0.07) |
| Organics | $\mu g/m^3$ | 0.45 (0.49) | 12.70 | 0.75 (1.08) | 0.07 (0.08) |
| Nitrate ($NO_3$) | $\mu g/m^3$ | 0.07 (0.11) | 0.58 | 0.03 (0.03) | 0.00 (0.00) |
| Sulphate ($SO_4$) | $\mu g/m^3$ | 0.11 (0.02) | 0.24 | 0.08 (0.02) | 0.12 (0.02) |
| Black carbon (BC) | $\mu g/m^3$ | 0.03 (0.03) | 1.39 | 0.04 (0.04) | 0.00 (0.00) |
| Scattering coefficient at 550 nm | $Mm^{-1}$ | 4.82 (4.68) | 366.41 | 14.30 (11.40) | 1.51 (1.19) |
| Absorption coefficient at 370 nm | $Mm^{-1}$ | 0.55 (0.75) | 53.78 | 1.76 (1.71) | 0.17 (0.38) |
| Absorption coefficient at 550 nm | $Mm^{-1}$ | 0.33 (0.37) | 18.84 | 0.53 (0.50) | 0.06 (0.04) |
| Absorption Ångström exponent (AAE) 370-880 nm | - | 1.12 (0.51) | 1.86 | 2.11 (0.81) | 2.34 (4.25) |
| Scattering Ångström exponent (SAE) 450-700 nm | - | 2.22 (0.49) | -0.11 | 0.30 (0.45) | 1.42 (1.37) |
| Single scattering albedo (SSA) at 550 nm | - | 0.95 (0.04) | 0.95 | 0.96 (0.01) | 0.97 (0.05) |
| SSA Ångström exponent (SSA_AE) 450-700 nm | - | 0.05 (0.05) | -0.13 | -0.08 (0.04) | 0.04 (0.20) |
| Number concentration | $1/cm^3$ | 394.84 (171.79) | 544.61 | 330.19 (141.69) | 324.37 (65.97) |
| Volume concentration | $\mu m^3/cm^3$ | 0.67 (0.64) | 33.39 | 1.37 (1.00) | 0.22 (0.17) |



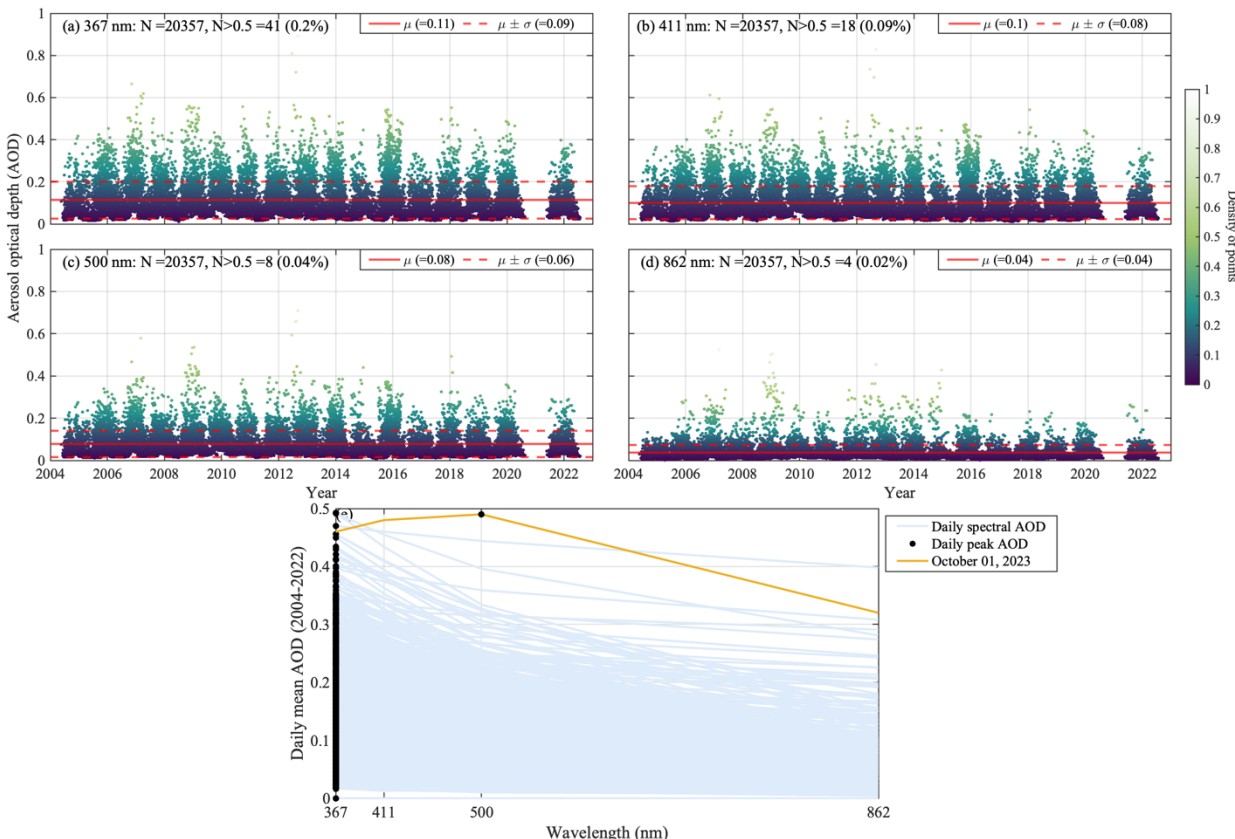


**Figure A1: (a-d) Climatological spectral AOD variation at 367, 411, 500 and 862 nm and, (e) Daily mean spectral AOD and (e) daily spectral maxima (black dots) between 2004-2022 from GAWPFR 01 October 2023 (yellow) at DAV**



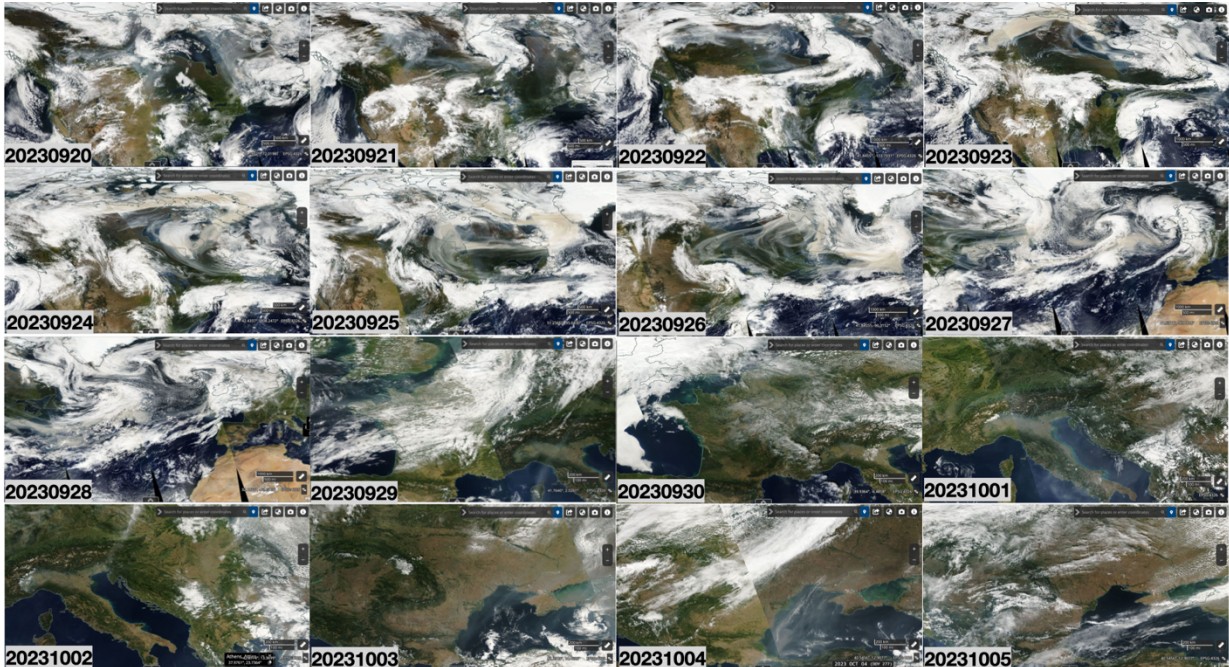

- Brown plumes moving out from Canada from 23rd September and reached Northern Europe around 27th September
- Smoke over Switzerland and Italy on 1st October

**Figure A2: MODIS true colour images tracing the plume. Imagery credits: © Land Atmosphere Near real-time Capability for EOS (LANCE) system and Global Imagery Browse Services (GIBS), operated by NASA Earth Observing System Data and Information System (EOSDIS).**

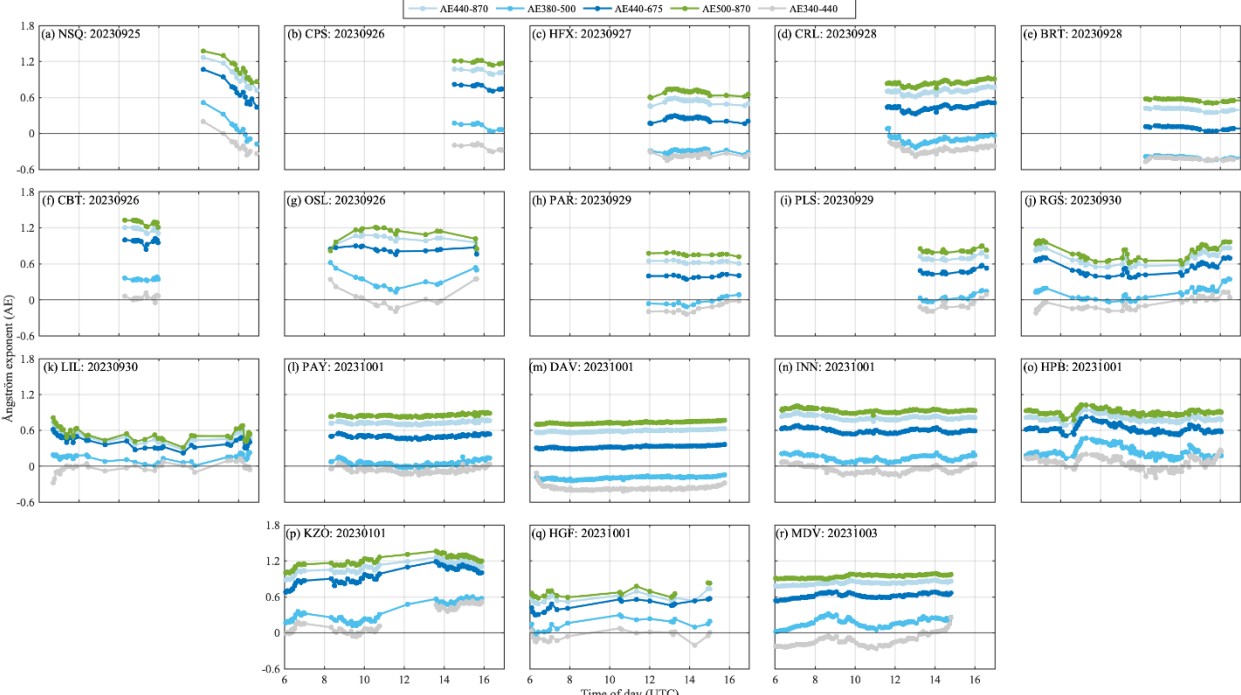



**Figure A3: (a-r) Variation of AE during the peak day of the event at the respective stations.**

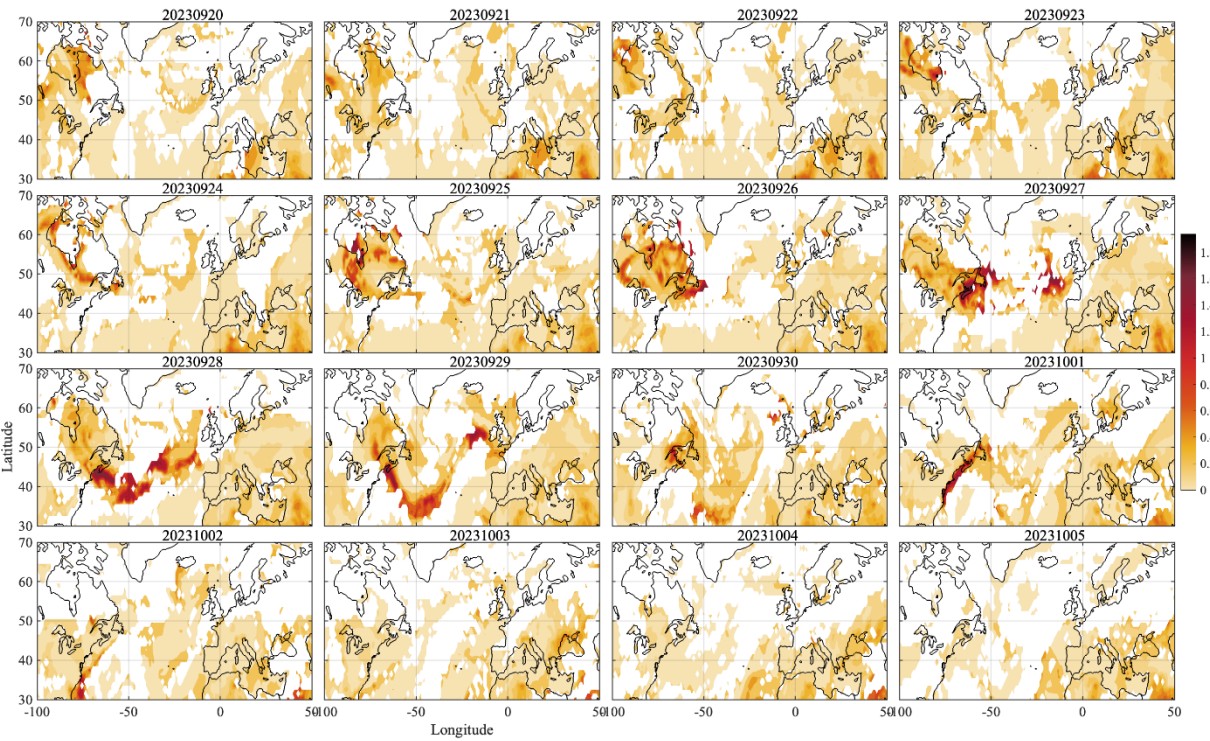

**Figure A4: MODIS AOD at 550 nm during 20 September and 05 October 2023.**





**Table A2: Correlation between mean size distribution of fine and coarse mode and peak in AOD concave spectral curvature during the peak day of the event at considered stations.**

| Station | Time (UTC) | Volume concentration (fine, coarse, total) | Volume median radius (fine, coarse, total) | Effective radius (fine, coarse, total) | Peak AOD wavelength |
|---------|------------|--------------------------------------------|--------------------------------------------|----------------------------------------|---------------------|
| | | Stations west of North Atlantic Ocean | | | |
| HFX | 14:09 | 0.23, 0.07, 0.29 | 0.33, 4.39, 0.59 | 0.31, 3.55, 0.38 | 500 |
| | 17:09 | 0.28, 0.07, 0.35 | 0.33, 4.65, 0.57 | 0.31, 3.77, 0.38 | 500 |
| | 18:09 | 0.31, 0.06, 0.36 | 0.33, 4.86, 0.50 | 0.31, 3.85, 0.36 | 500 |
| | 19:09 | 0.25, 0.06, 0.31 | 0.32, 4.74, 0.53 | 0.29, 3.86, 0.35 | 440 |
| CRL | 12:13 | 0.10, 0.03, 0.14 | 0.33, 3.93, 0.58 | 0.30, 3.18, 0.38 | 440 |
| | 12:52 | 0.13, 0.03, 0.15 | 0.32, 4.54, 0.52 | 0.29, 3.61, 0.35 | 500 |
| | 13:50 | 0.13, 0.03, 0.16 | 0.33, 3.96, 0.50 | 0.30, 3.07, 0.35 | 440 |
| | 14:50 | 0.14, 0.03, 0.17 | 0.32, 3.54, 0.50 | 0.29, 2.72, 0.35 | 440 |
| | 15:50 | 0.16, 0.03, 0.19 | 0.31, 3.90, 0.48 | 0.29, 3.05, 0.34 | 440 |
| | 16:50 | 0.18, 0.04, 0.21 | 0.31, 3.88, 0.48 | 0.28, 3.10, 0.34 | 440 |
| | 17:50 | 0.22, 0.05, 0.27 | 0.32, 4.51, 0.51 | 0.29, 3.56, 0.35 | 440 |
| | 18:50 | 0.20, 0.04, 0.24 | 0.32, 4.33, 0.48 | 0.29, 3.35, 0.34 | 440 |
| | 19:51 | 0.19, 0.04, 0.23 | 0.32, 4.30, 0.50 | 0.29, 3.42, 0.35 | 440 |
| BRT | 13:39 | 0.22, 0.05, 0.27 | 0.35, 5.23, 0.56 | 0.33, 4.25, 0.39 | 500 |
| | 14:39 | 0.24, 0.04, 0.28 | 0.36, 5.22, 0.54 | 0.33, 4.19, 0.39 | 500 |
| | 16:40 | 0.22, 0.04, 0.26 | 0.35, 4.72, 0.53 | 0.33, 3.72, 0.38 | 500 |
| | | Stations east of North Atlantic Ocean | | | |
| CBT | 11:08 | 0.05, 0.02, 0.07 | 0.27, 2.51, 0.49 | 0.24, 2.05, 0.32 | 380 |
| OSL | 11:19 | 0.06, 0.03, 0.09 | 0.27, 2.08, 0.52 | 0.25, 1.74, 0.34 | 380 |
| PLS | 13:52 | 0.05, 0.01, 0.06 | 0.32, 3.99, 0.48 | 0.28, 3.25, 0.33 | 440 |
| RGS | 11:45 | 0.06, 0.03, 0.09 | 0.34, 4.27, 0.89 | 0.31, 3.66, 0.48 | 440 |
| | 15:28 | 0.05, 0.02, 0.07 | 0.30, 3.23, 0.59 | 0.25, 2.62, 0.34 | 380 |
| DAV | 09:22 | 0.07, 0.00, 0.07 | 0.33, 5.24, 0.38 | 0.30, 3.74, 0.32 | 500 |
| | 10:22 | 0.06, 0.01, 0.07 | 0.33, 3.85, 0.39 | 0.30, 2.75, 0.32 | 500 |
| | 11:22 | 0.05, 0.00, 0.05 | 0.34, 4.98, 0.38 | 0.32, 3.50, 0.33 | 500 |
| | 12:22 | 0.05, 0.00, 0.05 | 0.34, 4.64, 0.39 | 0.32, 3.19, 0.33 | 500 |
| | 13:22 | 0.05, 0.00, 0.05 | 0.34, 5.49, 0.37 | 0.32, 3.84, 0.33 | 500 |
| | 14:22 | 0.06, 0.00, 0.06 | 0.35, 4.34, 0.42 | 0.33, 3.09, 0.35 | 500 |
| INN | 07:27 | 0.05, 0.01, 0.06 | 0.30, 3.26, 0.46 | 0.26, 2.61, 0.31 | 380 |
| | 10:15 | 0.04, 0.01, 0.04 | 0.31, 3.76, 0.42 | 0.28, 3.10, 0.32 | 440 |
| | 11:15 | 0.03, 0.01, 0.04 | 0.30, 3.05, 0.40 | 0.27, 2.45, 0.30 | 380 |
| | 12:15 | 0.03, 0.00, 0.04 | 0.30, 2.78, 0.39 | 0.27, 2.21, 0.30 | 440 |
| | 13:15 | 0.03, 0.01, 0.03 | 0.30, 3.10, 0.43 | 0.27, 2.52, 0.31 | 380 |
| | 14:15 | 0.03, 0.01, 0.03 | 0.30, 3.44, 0.46 | 0.28, 2.79, 0.33 | 380 |
| | 15:03 | 0.03, 0.01, 0.04 | 0.32, 4.25, 0.46 | 0.30, 3.49, 0.34 | 380 |
| | 15:28 | 0.03, 0.01, 0.04 | 0.32, 4.41, 0.49 | 0.30, 3.69, 0.34 | 380 |
| HPB | 09:17 | 0.02, 0.00, 0.02 | 0.31, 2.71, 0.49 | 0.27, 2.32, 0.34 | 380 |
| | 10:17 | 0.01, 0.00, 0.02 | 0.30, 2.76, 0.48 | 0.26, 2.32, 0.32 | 380 |
| | 11:17 | 0.01, 0.00, 0.02 | 0.30, 3.51, 0.40 | 0.26, 2.79, 0.29 | 380 |
| | 13:17 | 0.02, 0.00, 0.02 | 0.31, 2.57, 0.39 | 0.27, 2.14, 0.30 | 380 |
| | 14:17 | 0.01, 0.00, 0.01 | 0.31, 2.60, 0.48 | 0.28, 2.21, 0.34 | 380 |



| | 15:04 | 0.02, 0.00, 0.02 | 0.31, 4.33, 0.49 | 0.28, 3.58, 0.33 | 380 |
|---|---|---|---|---|---|
| | 15:28 | 0.02, 0.00, 0.02 | 0.32, 3.01, 0.45 | 0.30, 2.55, 0.34 | 380 |
| KZO | 09:56 | 0.02, 0.00, 0.02 | 0.26, 3.04, 0.33 | 0.23, 2.34, 0.25 | 380 |
| | 14:45 | 0.01, 0.00, 0.02 | 0.28, 3.13, 0.42 | 0.24, 2.56, 0.28 | 340 |
| | 15:18 | 0.01, 0.00, 0.01 | 0.28, 3.68, 0.43 | 0.24, 3.02, 0.29 | 340 |
| MDV | 05:38 | 0.04, 0.02, 0.06 | 0.32, 3.51, 0.71 | 0.30, 2.99, 0.43 | 440 |
| | 06:18 | 0.04, 0.02, 0.06 | 0.32, 3.49, 0.75 | 0.30, 2.94, 0.43 | 440 |
| | 07:05 | 0.03, 0.02, 0.05 | 0.28, 3.18, 0.76 | 0.25, 2.52, 0.39 | 440 |
| | 08:05 | 0.03, 0.03, 0.06 | 0.28, 3.30, 0.93 | 0.25, 2.77, 0.45 | 380 |
| | 09:05 | 0.03, 0.03, 0.06 | 0.29, 3.37, 0.88 | 0.25, 2.93, 0.44 | 380 |
| | 10:05 | 0.03, 0.02, 0.05 | 0.29, 3.31, 0.66 | 0.26, 2.83, 0.38 | 380 |
| | 11:05 | 0.04, 0.02, 0.06 | 0.29, 3.12, 0.68 | 0.27, 2.50, 0.39 | 440 |
| | 12:05 | 0.03, 0.03, 0.06 | 0.28, 3.49, 1.00 | 0.25, 2.81, 0.46 | 380 |
| | 13:05 | 0.03, 0.03, 0.06 | 0.28, 3.35, 0.98 | 0.26, 2.73, 0.47 | 380 |
| | 13:52 | 0.03, 0.03, 0.06 | 0.29, 3.53, 1.05 | 0.26, 2.90, 0.49 | 380 |
| | 14:14 | 0.03, 0.03, 0.06 | 0.30, 3.46, 1.05 | 0.27, 3.03, 0.50 | 380 |


**Table A3: Comparison of MODIS and MERRA2 AOD statistics with AEROENT measured AOD during the peak day of the event. The percentage differences are calculated based on AERONET AOD.**

| Station | Mean AOD Difference (MODIS-AERONET) | Mean AERONET AOD | Percentage difference (%) | Station | Mean AOD Difference (MERRA2-AERONET) | Mean AERONET AOD | Percentage difference (%) |
|---|---|---|---|---|---|---|---|
| Overestimation | | | | Overestimation | | | |
| PLS | 0.39 | 0.33 | 118.18 | KZO | 0.12 | 0.12 | 100.00 |
| PAR | 0.31 | 0.42 | 73.81 | HPB | 0.05 | 0.12 | 41.67 |
| HFX | 0.21 | 1.97 | 10.66 | RGS | 0 | 0.33 | 0.00 |
| CPS | 0.11 | 0.89 | 12.36 | Underestimation | | | |
| KZO | 0.04 | 0.12 | 33.33 | HFX | -1.51 | 1.97 | -76.65 |
| LIL | 0.05 | 0.22 | 22.73 | NSQ | -1.32 | 1.48 | -275.00 |
| Underestimation | | | | BRT | -1.25 | 1.46 | -85.62 |
| BRT | -0.29 | 1.46 | -19.86 | CRL | -0.94 | 1.22 | -77.05 |
| HGF | -0.28 | 0.49 | -57.14 | CPS | -0.79 | 0.89 | -88.76 |
| DAV | -0.25 | 0.46 | -54.35 | OSL | -0.38 | 0.52 | -73.08 |
| INN | -0.17 | 0.27 | -62.96 | DAV | -0.38 | 0.46 | -82.61 |
| PAY | -0.15 | 0.28 | -53.57 | HGF | -0.31 | 0.49 | -63.27 |
| MDV | -0.14 | 0.27 | -51.85 | CBT | -0.23 | 0.39 | -58.97 |
| CBT | -0.13 | 0.39 | -33.33 | MDV | -0.18 | 0.27 | -66.67 |
| CRL | -0.11 | 1.22 | -9.02 | PAY | -0.16 | 0.28 | -57.14 |
| RGS | -0.08 | 0.33 | -24.24 | PAR | -0.13 | 0.42 | -30.95 |
| HPB | -0.02 | 0.12 | -16.67 | INN | -0.13 | 0.27 | -48.15 |
| | | | | LIL | -0.03 | 0.22 | -13.64 |
| | | | | PLS | -0.02 | 0.33 | -6.06 |



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
