# Peer review of "Long range transport of Canadian Wildfire smoke to Europe in Fall 2023: aerosol properties and spectral features of smoke particles"

_EGUsphere, 2025_

## Author Comment (AC1)

**Response to community comments of Dr. Thomas Eck**

This is an interesting paper with useful observations of rare smoke aerosol characteristics. I have a few relatively minor comments with the most significant pertaining to the Abstract (see below).

**Comment #1:** Abstract: The 2023 biomass burning season in Canada exceeded 5 months duration, and the observations focused on in this manuscript occurred during a one-week interval at the end of this very long biomass burning season.  It needs to be emphasized that the other 5 months of smoke observations were very different from these and that makes it quite notable. Additionally, the word 'unique' occurs twice in this abstract and it should be replaced by 'very rare' in both instances.  The authors have made it very clear in the Conclusions section that these aerosol characteristics are not unique since other observations of these very unusual AOD spectra (maximum at 500 nm), size distributions and spectral single scattering albedo characteristics were documented by Eck et al. (2023) from forest fire smoke originating in California and Oregon in September 2020.

**Response #1:** We are thankful for the community comment.

We agree the use of the word "unique" is not accurate in this case and hence have replaced the word "unique" with "rare" throughout the manuscript.

We have added a line in the abstract as,

"The Canadian wildfires of 2023 had an unprecedented biomass burning season spanning from mid-April to late October. Towards the end of this long biomass burning season, there was a rare observation of smoke properties that occurred for about a week interval, that were different from the whole biomass burning season which makes it quite notable."

Accordingly, we have updated the title as,

"Long range transport of Canadian Wildfire smoke to Europe in 2023: aerosol properties and spectral features of smoke particles"

as the observations presented in this manuscript are only for about a week and are not representative of the entire fall season.

**Comment #2:** Lines 136-137: Since the calibration uncertainty of the PFRs AOD are given in the paragraph below the uncertainty of the AERONET AOD data should also be mentioned here.  The AERONET field instruments are inter-calibrated versus Mauna Loa and Izana Langley calibrated reference instruments resulting in AOD uncertainty at optical airmass 1 of ~0.01 in the visible and near infrared increasing to ~0.02 in the UV wavelengths. (Eck et al. (1999)).

**Response #2:** We are thankful for this suggestion following which we have added the following line in the updated manuscript in Section 2.1.1.

"The AERONET field instruments are inter-calibrated at Mauna Loa and Izana Langley calibrated reference instruments resulting in AOD uncertainty at optical airmass 1 of ~0.01 in the visible and near infrared increasing to ~0.02 in the UV wavelengths (Eck et al. 1999)."

**Comment #3:** Lines 490-492: This is incorrect. It seems that 'AOD' should have been 'SSA' since the topic of this section and sentences is spectral absorption, and therefore spectral AOD makes little sense here.

**Response #3:** We are thankful for this correction and agree with this change.

We have changed the "spectral AOD variation" to "spectral SSA variation".

**Comment #4:** Line 609: This sentence is also incorrect. Eck et al (2023) attributed the concave spectral shape of AOD curvature to the large size radius of sub-micron particles and narrow width of this fine mode distribution. This was even modeled by Mie simulations by Eck et al (2023), see Figure 12. Maybe you mistakenly meant concave spectral "SSA" spectral curvature instead of "AOD" curvature here?

**Response #4:** We are thankful for this correction.

We have replaced the Line "Similar concave spectral AOD curvature was observed in another wildfire event in California/Oregon in 2020 as presented by Eck et al. (2023) in which the authors suggested the presence of coated black carbon and/or BrC" with the lines as below,

"Similar concave spectral AOD curvature was observed in another wildfire event in California/Oregon in 2020 as presented by Eck et al (2023) in which the authors attributed the concave spectral shape of AOD curvature to the large size radius of sub-micron particles and narrow width of this fine mode distribution."

**Comment #5:** Lines 611-612: This switching back to AOD spectral curvature while discussing SSA spectral curvature in this short paragraph is confusing. Please consider revising in order to clarify here.

**Response #5:** We are thankful for this suggestion. We have updated this paragraph as below,

"Similar concave spectral AOD curvature was observed in another wildfire event in California/Oregon in 2020 as presented by Eck et al. (2023) in which the authors attributed the concave spectral shape of AOD curvature to the large size radius of sub-micron particles and narrow width of the fine mode size distribution. A Canada wildfire transport to Europe in 1950 had the observation of the solar spectrum extinction minima at 4350 Å (435 nm) as obtained from solar spectrograms in Edinburgh in September, during which, there was observation of blue sun as presented by Wilson 1951."

**Comment #6:** Line 613: I suggest adding something like: "plus Canada forest fire smoke transported to Scotland in September 1950 (Wilson, 1951)."

**Response #6:** We are thankful for this suggestion following which we have added the lines as below in the conclusion section,

"A Canada wildfire transport to Europe in 1950 had the observation of the solar spectrum extinction minima at 4350 Å (435 nm) as obtained from solar spectrograms in Edinburgh in September, during which, there was observation of blue sun as presented by Wilson 1951."

References:

Eck, T. F., B. N. Holben, J. S. Reid, O. Dubovik, A. Smirnov, N. T. O'Neill, I. Slutsker, and S. Kinne (1999), Wavelength dependence of the optical depth of biomass burning, urban, and desert dust aerosols, J. Geophys. Res., 104(D24), 31333–31349, doi:10.1029/1999JD900923.

Wilson, R.: The blue sun of 1950 September. Mon. Not. Roy. Astron. Soc. 111 (5), 478–489, doi:10.1093/mnras/111.5.478, 1951.

We are thankful to Dr. Thomas Eck for the suggestions and comments that we have incorporated in the current version of the manuscript.